# Molecular basis of differential HLA class I-restricted T cell recognition of a highly networked HIV peptide

Xiaolong Li [1,2,3] ✉, Nishant Kumar Singh[2,4,5], David R. Collins [2,5], Robert Ng[3], Angela Zhang[2,3], Pedro A. Lamothe-Molina [2], Peter Shahinian [2], Shutong Xu [3], Kemin Tan [6], Alicja Piechocka-Trocha[2,5], Jonathan M. Urbach [2], Jeffrey K. Weber[7], Gaurav D. Gaiha[2,8], Overbeck Christian Takou Mbah[2], Tien Huynh[7], Sophia Cheever[2], James Chen [2], Michael Birnbaum [2,4], Ruhong Zhou [7,9,10], Bruce D. Walker [2,5,11] ✉ & Jia-huai Wang [3,12,13] ✉

Cytotoxic-T-lymphocyte (CTL) mediated control of HIV-1 is enhanced by targeting highly networked epitopes in complex with human-leukocyte-antigen-class-I (HLA-I). However, the extent to which the presenting *HLA* allele contributes to this process is unknown. Here we examine the CTL response to QW9, a highly networked epitope presented by the disease-protective HLA-B57 and disease-neutral HLA-B53. Despite robust targeting of QW9 in persons expressing either allele, T cell receptor (TCR) cross-recognition of the naturally occurring variant QW9_S3T is consistently reduced when presented by HLA-B53 but not by HLA-B57. Crystal structures show substantial conformational changes from QW9-HLA to QW9_S3T-HLA by both alleles. The TCR-QW9-B53 ternary complex structure manifests how the QW9-B53 can elicit effective CTLs and suggests sterically hindered cross-recognition by QW9_S3T-B53. We observe populations of cross-reactive TCRs for B57, but not B53 and also find greater peptide-HLA stability for B57 in comparison to B53. These data demonstrate differential impacts of HLAs on TCR cross-recognition and antigen presentation of a naturally arising variant, with important implications for vaccine design.

The past several decades have seen the emergence of multiple viruses with pandemic potential, including HIV, Ebola, Zika, SARS, MERS, SARS-CoV-2 and most recently monkeypox. A major obstacle to immune control of viral infections is the ability of viruses to rapidly mutate and evade both humoral and cellular host immune responses[1,2]. However, at least in the case of HIV-1 infection, a small percentage of infected persons (less than 0.5%) maintain spontaneous viral control without treatment (termed "HIV controllers", as opposed to "progressors"), suggesting an inherent host capability for induction of effective immunity[1].

This kind of HIV protection, *albeit* rare, represents a natural model for immune-mediated control of a human viral infection. Genome-wide association studies (GWAS) have provided compelling statistical evidence indicating that the *human-leukocyte-antigen-class-I* (*HLA-I*) genotypes are strongly associated with disease outcomes in persons with HIV. In particular, *HLA-B* alleles such as *HLA-B*5701* and *HLA-B*2705*, mediate strong genetic influence on immune control of HIV in untreated disease, whereas other genotypes are associated with the progression to acquired immune deficiency syndrome (AIDS)[3,4]. Further, durable spontaneous control of HIV infection without

medications is associated with specific amino acids lining the HLA-I peptide binding groove involved in presenting viral peptides on infected cells for recognition by cytotoxic T lymphocytes (CTLs)[4], implicating the peptide-HLA (pHLA) complex in modulating disease outcome.

Recent studies applying network theory to HIV proteins indicate that viral control is also associated with the recognition of epitopes composed of structurally constrained amino acid residues (i.e. highly networked) by CTLs[5]. Highly networked epitopes contain amino acid residues involved in critical noncovalent interactions within the protein that maintain structural integrity and are thus much less tolerant of mutation due to potential effects on viral fitness. Notably, recognition of such highly networked epitopes has been shown to be associated with durable HIV control, even in the absence of protective *HLA* alleles[5]. One such highly networked epitope associated with control is QW9 (QASQEVKNW, HIV-1 p24gag residues 176-184). QW9 can be presented by the disease-protective HLA-B*5701 (hereafter B57), but also by the typically disease-neutral HLA-B*5301 (hereafter B53)[6].

Here we exploit this finding that a single highly networked epitope can be recognized in the context of two different *HLA* alleles to explore the extent to which the presenting HLA molecule affects CTL cross-recognition of epitope variants. We show that CTLs derived from B53-expressing (B53+) and B57-expressing (B57+) infected persons at a time of viremic control readily recognize the QW9 peptide. However, recognition of the naturally occurring variant QW9_S3T with a serine to threonine mutation at peptide position 3 (p3) differs substantially, with consistent reduction in recognition when the epitope is presented by B53. Through functional and structural studies coupled with biophysical measurements, we provide evidence for multifactorial mechanistic underpinnings of differential CTL recognition due to specific HLA amino acid residues.

## Results

### Differential cross-recognition of a naturally occurring QW9 variant by QW9-specific B53 and B57 restricted T cells

The p24 Gag capsid protein is statistically the most highly networked HIV protein[5] and contains the highly networked peptide QW9 (QAS-QEVKNW, HIV-1 p24gag residues 176–184) that encompasses multiple highly networked amino acid residues (Supplementary Table 1) which have been shown to have an important function in maintaining the integrity of p24, including V6QW9 at a typical T cell receptor contact site and the C terminal anchor residue W9QW9 (Supplementary Fig. 1a).

Interestingly, QW9 can be presented by both B53 and B57 (ref. 6). These two molecules are encoded by distinct *HLA* supertypes that differ in residues lining the antigen-binding groove, including HLA residue 97 on the β5 strand known to have important conformational properties for peptide binding[7] and multiple residues in the α1 helix having potential contact with TCR and/or peptide (Fig. 1a). These HLA residues have been shown to be associated with differential viral control via GWAS[4]. The two alleles also differ in their peptide binding specificity, with B53 preferring a proline at the position p2 anchor residue, whereas B57 prefers a serine or threonine[8,9].

To understand whether there is differential recognition of this highly networked epitope presented by protective and non-protective HLAs, we first analyzed responses in HIV-1 infected persons expressing B53 or B57. We identified five B53+ and five B57+ persons with HIV, all studied at a time point when viral load was relatively controlled without medications and CD4+ T cell counts were similar (Supplementary Table 2). Though QW9 can be presented by both B57 and B53, our flow cytometry results (Supplementary Fig. 2) showed that HLA-I tetramers constructed with respective *HLA* alleles and loaded with the QW9 peptide only reacted with T cells from persons expressing the matched HLA molecule, with no evidence of cross-recognition between alleles (Fig. 1b, c), indicating a dominant influence of *HLA* genotype on TCR recognition of the pHLA complex.

Despite similar recognition of QW9, QW9-HLA-tetramer-specific CTL lines generated from the B53+ cohort but not the B57+ cohort had a diminished ability to lyse target cells loaded with a naturally occurring viral mutant QW9_S3T. At saturating peptide concentrations, the functional capacity of QW9-specific CTLs to kill cells loaded with QW9_S3T was consistently less than cells loaded with QW9 in the context of B53 (Fig. 1d, P-value: 0.0028). In contrast, QW9-specific CTL lines from B57+ persons maintained cross-recognition and killing of autologous cells loaded with the QW9_S3T variant (Fig. 1e, P-value: 0.5408).

We also assessed CD8+ T cell proliferation in response to the recognition of targeted peptides, as this function has been shown to correlate with effective HIV control[10–13]. When stimulated with the variant QW9_S3T, we found that wild-type QW9-HLA-tetramer-specific CTLs from B53+ persons proliferated less than after exposure to the wild-type QW9 (Fig. 1f, P-value: 0.0227), while there was no significant difference for B57+ persons (Fig. 1g, P-value: 0.8200), consistent with the killing assays. Together, these functional data indicate that despite recognition of a highly networked epitope presented by B57 and B53, QW9-specific CTL cross-recognition of a naturally occurring mutant QW9_S3T is diminished in the context of B53, not B57.

### T cell cross-reactivity is associated with differential TCR recognition

To further examine the differential response of the QW9_S3T variant by CTLs from B53+ and B57+ individuals, we evaluated QW9- and QW9_S3T-specific CTL recognition using pHLA tetramers by flow cytometry. To examine whether cross-reactivity is mediated through CTLs with dual reactivity or by distinct CTL clonotypes recognizing either the wild-type or mutant peptide, we generated allophycocyanin (APC)-conjugated QW9-B53 and QW9-B57 tetramers, and phycoerythrin (PE)-conjugated QW9_S3T-B53 and QW9_S3T-B57 tetramers and used these for ex vivo staining of a B53+ and a B57+ PBMC. Consistent with ELISpot data (Fig. 2a, b), we observed specific HLA-restricted staining of QW9-specific and QW9_S3T-specific CD8+ T cells for both individuals (Fig. 2c, d, left and middle panels). However, only the B57+ individual showed a tetramer staining pattern consistent with cross-recognition of QW9 by a subset of QW9_S3T specific CTLs (Fig. 2c, right panel, highlighted in green), whereas the B53+ individual showed staining patterns consistent with distinct subsets of CTL clonotypes recognizing either QW9 or QW9_S3T with little or no cross-reactivity (Fig. 2d, right panel). We then used tetramers to isolate QW9-specific, QW9_S3T-specific and dual-reactive T cells from these persons. Single-cell TCR sequencing indicated that single-tetramer or dual-tetramer-stained cells expressed distinctive TCRs (Fig. 2e, f), with a population that was clearly cross-reactive in the B57 expressing individual and in one of two additional B57 participants tested, whereas no such cross-reactive cells were assuredly detected in the B53 expressing individual or in two additional B53 expressing persons (Supplementary Fig. 3). These data suggest one difference in cross recognition of the variant peptide is that some clonotypes of B57-restricted CTL responses cross-recognize QW9 and QW9_S3T, whereas we were unable to confirm such dual recognition in B53 expressing persons.

### QW9 is presented by B53 and B57 with an identical main-chain conformation

To explore the molecular underpinnings of differential cross-recognition of QW9 and QW9_S3T restricted by the two HLAs, we next determined the crystal structures of QW9-B53, QW9_S3T-B53, QW9-B57 and QW9_S3T-B57 (Supplementary Table 3). As the electron density maps show, every individual side chain of QW9 is well-defined in the high-resolution structures for analyses (Supplementary Fig. 1b–e).

Figure 3a shows the overall structure of QW9-B53, and Fig. 3b gives the detailed interactions between QW9 and B53. Several findings are worth noting: a) The serine residue at position 116 of B53 (S116B53)

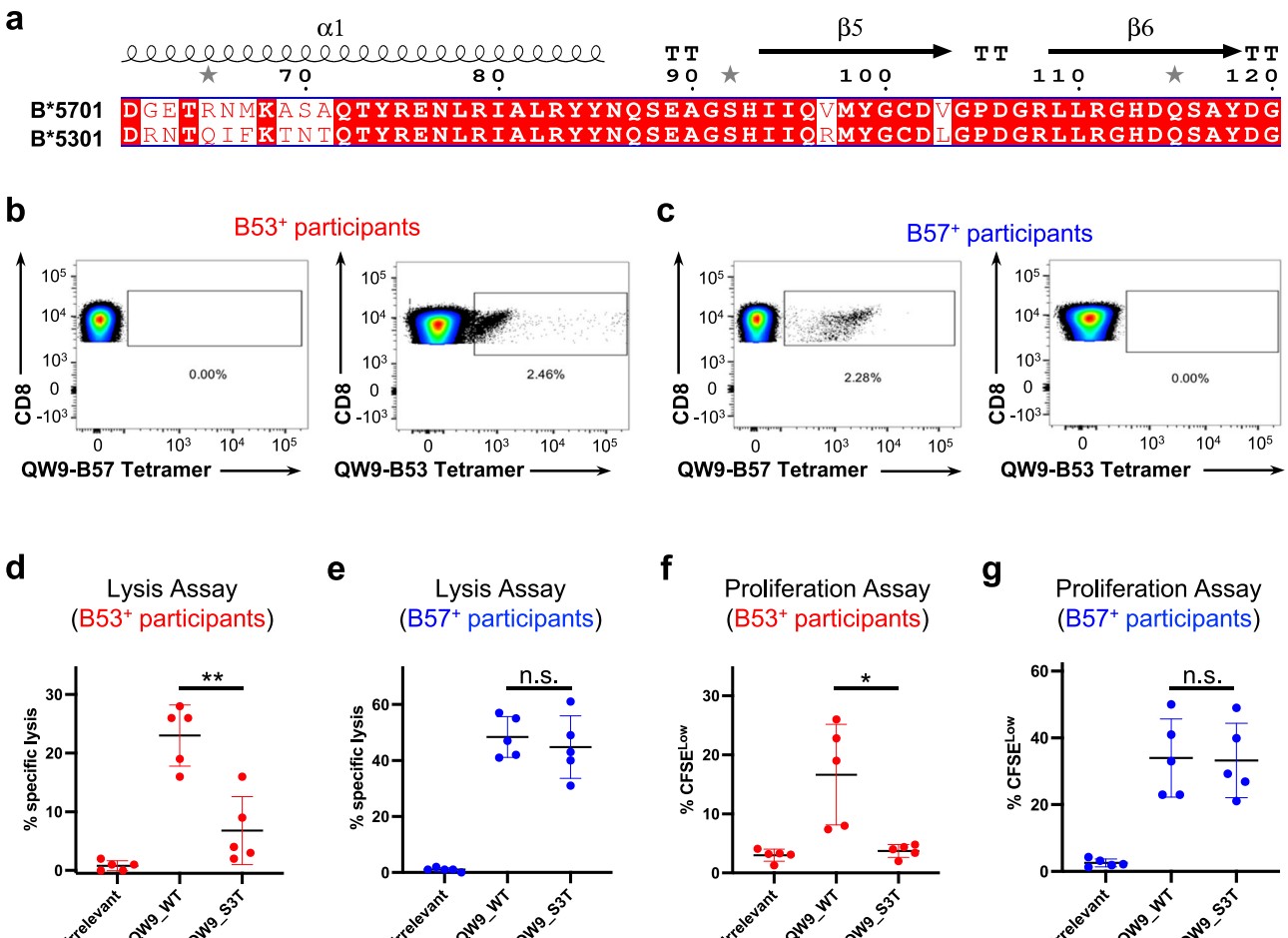

**Fig. 1 | Differential HLA-restricted immune recognition of a highly networked viral peptide. a** Sequence alignment for antigen binding domains of B53 and B57 (amino acid residues 60–120). The secondary structures are labeled at the top of the aligned sequences. **b, c** Flow cytometry plots of representative examples of QW9-HLA-tetramer-positive cells in HIV-infected persons expressing either B*5301 or B*5701. Each sample is dual stained with QW9-B53-APC and QW9-B57-PE tetramers. Systems were gated on CD3 + CD8+ live lymphocyte singlet cells. **d–g** Each data point represents a single person (Red, B*5301 persons; Blue: B*5701 persons) and each experiment contains $n = 5$ biologically independent samples. **d, e** Summarized data assessing specific lysis of target cells by QW9-HLA-specific T

cell lines in a standard 6 h chromium release assay at effector cell/target cell ratios of 1:1. Autologous EBV-transformed BCLs pulsed with the QW9 and QW9_S3T peptides were used as target cells (*P*-values: panel **d**, 0.0028; panel **e**, 0.5408). **f, g** Quantification of antigen-specific cell proliferation (CFSE$^{Low}$ cells). Cells were simulated for 6 days with either irrelevant peptides, the QW9 peptide, or the QW9_S3T mutant, and CD8+ CFSE low cells counted (*P*-values: **f**, 0.0227; **g**, 0.8200). **d–g** Two-tailed paired *t* test and 95% confidence interval were used for *P*-values calculation, n.s. $p > 0.05$, *$p \leq 0.05$, **$p \leq 0.01$, ***$p \leq 0.001$. The red or blue Error bars represent SD, and the black bars at the center represent the mean values.

has a small sidechain that creates a spacious F pocket[14] to accommodate the large aromatic sidechain of W9$^{QW9}$ which stacks against the phenol ring of Y123$^{B53}$ and also forms a hydrogen bond to S116$^{B53}$ through a water molecule, thereby enhancing the peptide-HLA interaction at this primary anchor site; b) Residue R97$^{B53}$, the most frequently existing amino acid at position 97 among *HLA* alleles[4], sits at the central bottom of the B53 binding groove, connecting to its surroundings with five hydrogen bonds, including two to D114$^{B53}$ (Fig. 3b). These bonds should effectively neutralize the positively charged guanidinium moiety of the R97$^{B53}$ side chain, increasing stability as physiochemically suggested;[15] c) QW9 residues Q4$^{QW9}$, V6$^{QW9}$ and N8$^{QW9}$ prominently protrude out of the B53 groove (Fig. 3a, b), offering potential TCR-binding sites; d) Outward residue Q1$^{QW9}$ forms hydrogen bonds with R62$^{B53}$ and N63$^{B53}$ (Fig. 3b), making this amino acid less accessible to engage the TCR. These observations all suggest that QW9 can be loaded effectively on disease-neutral B53 for TCR recognition.

Despite HLA sequence differences within the binding groove and differences in TCR recognition, the structures of QW9-B57 and QW9-B53 can be superimposed remarkably well (Fig. 3c). Only two notable disparities exist in the conformations of QW9 bound to B53 and B57, at

residues Q1$^{QW9}$ and E5$^{QW9}$. In QW9-B53, E5$^{QW9}$ is oriented inward by HLA residue R97$^{B53}$ through S3$^{QW9}$ and a water-mediated hydrogen-bond network (Fig. 3b). By contrast, for B57, a hydrophobic V97$^{B57}$ takes the place of the R97$^{B53}$, and the negatively charged E5$^{QW9}$ hence points outward (Fig. 3c). Also, in QW9-B57, Q1$^{QW9}$ projects outward toward the putative TCR interface (Fig. 3c), owing to the replacement of the R62$^{B53}$-Q65$^{B53}$ by the G62$^{B57}$-R65$^{B57}$. These differential residues located at the HLA α1 helix also generate a unique electrostatic potential (ESP) surface for each HLA (Fig. 3d). These results provide structural evidence supporting the influence of *HLA* genotype on TCR recognition observed in the tetramer staining experiments (Fig. 1b, c), suggesting differential TCR engagement on the two pHLAs.

## The S3T mutation of QW9 induces the same substantial conformational changes when bound to B53 and B57

The substitution of serine with threonine in the QW9_S3T mutant provides an extra methyl group that is absent in the QW9, resulting in a substantial change in peptide presentation by B53. The changes are not caused by crystal packing (Supplementary Fig. 4a). Figure 3e overlays the structure of QW9_S3T-B53 with that of QW9-B53. The biggest

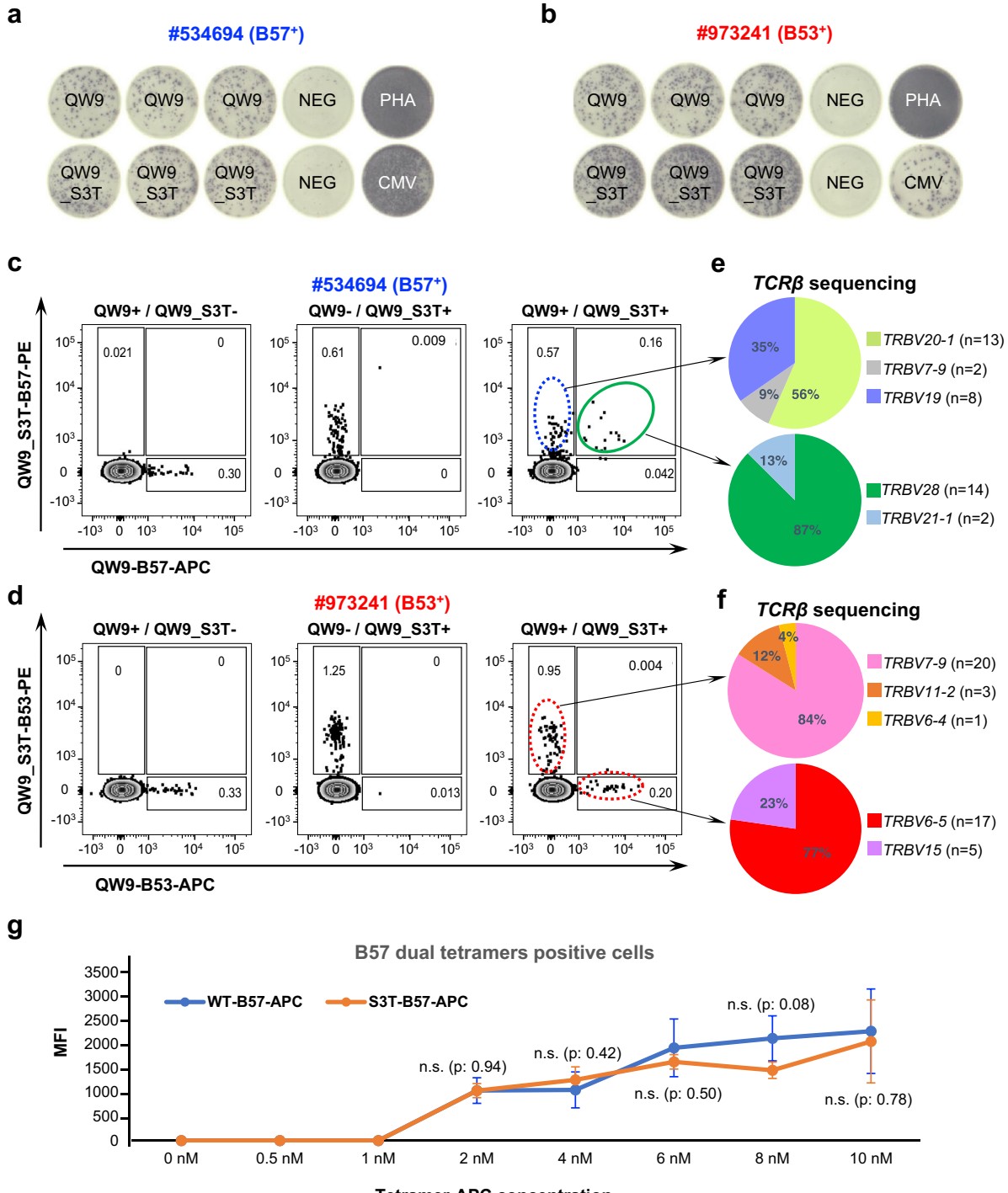

**Fig. 2 | Differential cross-reactivity of QW9-specific CTLs in B53⁺ and B57⁺ individuals. a, b** B57⁺ and B53⁺ individuals having responses to both QW9 and QW9_S3T, examined by ELISpot assay. NEG: no peptide (negative control); PHA: Phytohaemagglutinin (positive control); CMV: cytomegalovirus peptides (positive control). **c, d** Tetramer-specific CTL quantitation by flow cytometry. Columns left to right show QW9 tetramer, QW9_S3T tetramer, and dual tetramer staining. Rows depict individual participants expressing B57 or B53, as indicated. Dual QW9⁺/QW9_S3T⁺ tetramer staining population from a B57⁺ individual is highlighted in the green circle. **e, f** TCRβ gene usage in each CTL subset analyzed from single-cell TCR sequencing. N represents the number of cells in each clonotype. **g** Median Fluorescence Intensity (MFI) of QW9_B57-APC (blue curve) and QW9_S3T-B57-APC (orange curve) tetramer staining of cross-reactive, dual tetramer-positive T cells across a range of tetramer concentrations. The error bars represent standard deviation, duplicate number: *n* = 3. The difference of MFI between the QW9_B57-APC and QW9_S3T-B57-APC-stained cells at each concentration was labeled. Two-tailed unpaired *t* test and 95% confidence interval were used for *P*-values calculation (n.s: *p* > 0.05).

change is QW9 residue K7$^{QW9}$ flipping from being buried in QW9-B53 to being exposed in QW9_S3T-B53. Since the position of HLA residue Y99$^{B53}$ is fixed by a hydrogen bond to the amide group of T3$^{QW9\_S3T}$, the distances from the OH group of the Y99$^{B53}$ sidechain to Cα, Cβ and Cγ

atoms of T3$^{QW9\_S3T}$ are only 4.1 Å, 4.4 Å and 3.8 Å, respectively (Fig. 3e). It is thus reasonable to postulate that the bulky HLA residue Y99$^{B53}$ pushes the sidechain of the bound mutant peptide T3$^{QW9\_S3T}$ upward, consequently forcing E5$^{QW9\_S3T}$ to rotate away (Fig. 3e). These

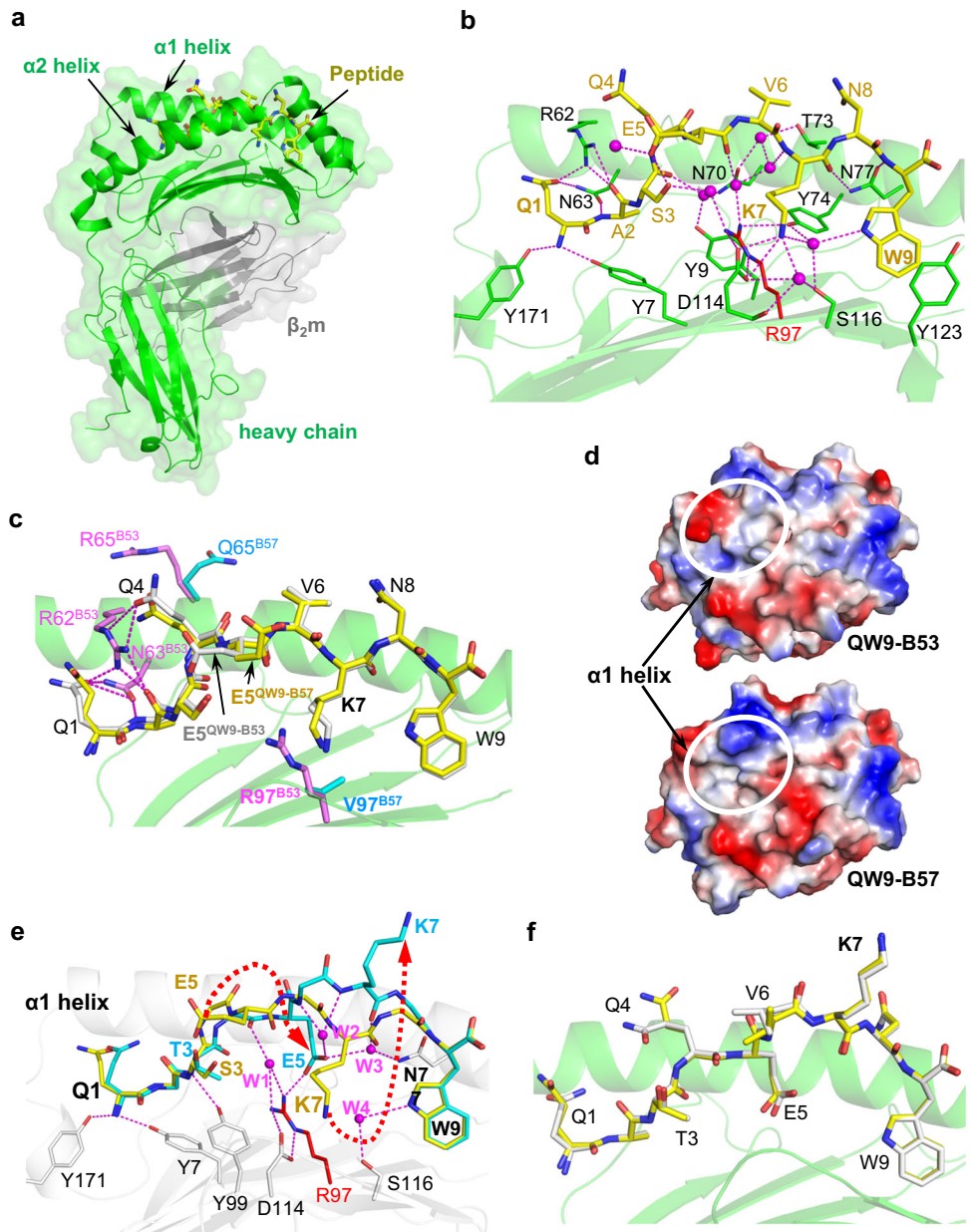

**Fig. 3 | Crystal structures of QW9 and QW9_S3T complexed with B53 and B57.**
**a** Overview of the QW9-B53 structure. The green and gray ribbons represent B53 heavy and light chains, respectively, with a transparent surface as the background. The QW9 peptide is shown as a yellow stick. **b** The detailed interaction between B53 and QW9. For clarity, the α2 helix of B53 is removed. Water molecules: magenta spheres; hydrogen bonds: magenta broken lines. B53 is represented as a green ribbon and QW9 as a yellow stick. Specific residues from B53 are shown as green, except for residue 97 which is shown as a red stick. **c** Superimposition of QW9-B57 onto QW9-B53. (Silver stick: QW9$^{B53}$; yellow stick: QW9$^{B57}$; magenta sticks: B53

residues; cyan sticks: B57 residues; magenta broken lines: hydrogen bonds). **d** Top view representation of the electrostatic potential surface of QW9-B53 and QW9-B57. Significant differences between QW9-B53 and QW9-B57 are highlighted by white circles. **e** Overlay of QW9_S3T$^{B53}$ (cyan stick) onto QW9$^{B53}$ (yellow stick) in the context of the same B53 (white ribbon). Only key hydrogen bonds (magenta broken lines) and water molecules (magenta spheres) involved in the interaction between QW9_S3T and B53 are shown. **f** Superimposition of QW9_S3T-B57 onto QW9_S3T-B53 (silver stick: QW9_S3T$^{B53}$; yellow stick: QW9_S3T$^{B57}$).

conformational changes of both T3 and E5 likely cause a conformational alteration of the peptide backbone, propagated through K7$^{QW9\_S3T}$ to make the whole peptide much more protruded in QW9_S3T-B53, compared to QW9-B53 (Fig. 3e). The carboxyl group of E5$^{QW9\_S3T}$ is stabilized by a salt-bridge to R97$^{B53}$ and hydrogen bonds to two water molecules, W2 and W3, while the flipped K7$^{QW9\_S3T}$ is stabilized by extensive hydrophobic contacts from the α2 helix of B53 to K7$^{QW9\_S3T}$ (Supplementary Fig. 4b). The electron density map clearly defines the protruding sidechain of K7$^{QW9\_S3T}$ (Supplementary Fig. 1c).

The S3T mutation also induces the same conformational changes for the presentation by B57 (Fig. 3f), following the same way of

propagation (Supplementary Fig. 4c). However, without an arginine at position-97 of B57, two more water molecules, W5 and W6, mediate a hydrogen-bond network to stabilize the E5$^{QW9\_S3T}$ buried in the antigen binding groove of B57 (Supplementary Fig. 4c). In summary, despite differential residues in the binding groove of the two alleles, the structural data indicate that the S3T mutation induces a similar peptide conformation change in B57 and B53. Therefore, differential cross-recognition of QW9_S3T by B53- and B57-restricted QW9-specific CTLs is unlikely to result from structural differences in the peptide presentation but likely from structural differences in the HLAs which select distinct TCRs.

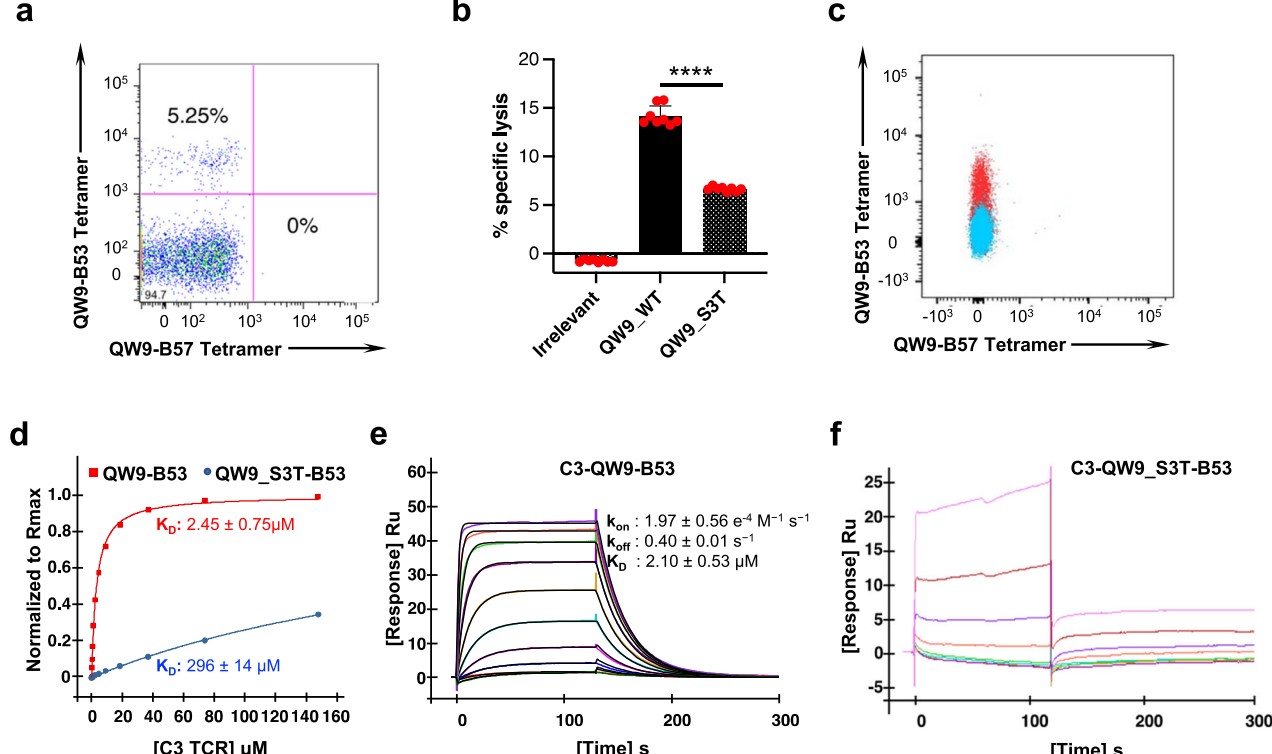

**Fig. 4 | QW9-specific, B53-restricted CTL isolation and functional examination.**
**a** PBMCs from a B53⁺ individual (viral load <50 RNA copies/mL, CD4 count = 1003, antiretroviral therapy-naive) were stimulated with QW9 peptide for 6 days and stained with an QW9-B53 tetramer. Expanded cells were subjected to single-cell FACS sorting by using QW9-B53 tetramers. **b** QW9-B53 specific CD8⁺ T cell, Clone 3, were expanded and tested for lysis of autologous B cells loaded with irrelevant, QW9 and QW9_S3T at an effector cell/target cell ratio of 1:1. Each peptide was tested with 8 technical replicates. Two-tailed unpaired *t* test and 95% confidence interval were used for *P*-value calculation, ****$p \leq 0.0001$. The Error bars represent SD. **c** The C3 TCR was cloned into a lentivirus and used to transduce a TCR null Jurkat cell line. C3 TCR expression was confirmed by flow cytometry as evidenced by binding to QW9-B53-APC tetramer staining (red cloud) in comparison to mock transduced

cells (blue cloud). *X*-axis: QW9-B57-PE conjugate, *Y*-axis: QW9-B53-APC conjugate. **d** Representative steady state SPR measurements for C3 interacting with QW9-B53 and QW9_S3T-B53. C3 TCR flowed through a QW9-B53 and QW9_S3T-B53 coated chip (the red curve represents QW9-B53, and the blue curve represents QW9_S3T-B53; duplicate number, *n* = 2). **e, f** Representative kinetic state SPR measurements for C3 interacting with QW9-B53 (panel **e**) and QW9_S3T-B53 (panel **f**) at 25 °C. 10 serial dilutions of concentrated TCR were injected at 50 μl/min throughout the experiment, with a contact time of 120 s and a dissociation time of 300 s (duplicate number, *n* = 2). Each binding response was fit to a 1:1 binding model, and the kinetic parameters were calculated using the BiaEval software, with $k_{on}$, $k_{off}$ and the $R_{max}$ fit globally.

## Highly networked epitope QW9 presented by the disease-neutral allele B53 can elicit an effective CTL response

We next sought to investigate the molecular features accounting for the recognition patterns observed. We were successful in generating a QW9-B53-specific CTL clone by single-cell fluorescence-activated cell sorting (FACS) following peptide stimulation for 6 days (Fig. 4a). The CTL clone, C3, lysed EBV-transformed autologous B-cell lines expressing QW9 epitope and its variant in a chromium release assay (Fig. 4b). We sequenced the TCR and transduced the C3 TCR into TCR-null Jurkat cells which were generated with CRISPR-mediated genetic knockout of *TCRα* and *TCRβ* genes (see methods), allowing us to evaluate the specific binding of the C3 TCR with QW9-B53 (Fig. 4c). We produced the soluble C3 TCR via a refolding method[16] and measured its binding affinity with wild-type QW9-B53 and the naturally arising mutant QW9_S3T-B53 using surface plasmon resonance (SPR). The affinity between the C3 TCR and QW9-B53 was 2.45 μM when measured at steady state, but there was a 120-fold decrease in the affinity of the C3 TCR to the mutant QW9_S3T when presented by B53 (Fig. 4d). Thus far, attempts to generate a QW9-B57 specific CTL clone for similar biochemical and structural TCR analysis have been unsuccessful. However, QW9-B57 and QW9_S3T-B57 dual-tetramer staining of cross-reactive CTLs demonstrated these B57 restricted TCRs have comparable binding affinity with wild-type QW9 and QW9_S3T bound B57, across a range of tetramer concentrations (Fig. 2g and Supplementary Fig. 5).

We also conducted binding kinetics measurements for the C3 TCR with QW9-B53 using SPR (Fig. 4e). The affinity of C3 TCR-QW9-B53 is at the high-affinity end of the 1–100 μM range for known TCR-pMHC-I interactions[17], indicating that C3 TCR indeed efficiently binds the QW9-B53 complex. The on-rate and off-rate also fall within the average for functional TCR-pHLA-I binding[18]. However, the kinetics of C3 binding the mutant peptide QW9_S3T-B53 appeared with rapid $k_{on}$ and $k_{off}$, too fast to measure using SPR (Fig. 4f), consistent with very weak binding between the C3 TCR and QW9_S3T-B53 shown in Fig. 4d. These data confirm at a clonal and structural level that it is possible to induce a high-affinity CTL response to the networked epitope QW9 in the context of the disease-neutral *HLA-B53* allele, but a single amino acid mutation in a naturally occurring HIV variant can significantly impair the established TCR recognition.

To further evaluate the C3 interaction with QW9-B53, we next solved crystal structures of the C3 TCR in complex with QW9-B53 (Fig. 5a) and, for comparison, the C3 TCR alone (Fig. 5b) at resolutions of 2.95 Å and 2.3 Å, respectively (Supplementary Table 3). The C3 TCR docks onto QW9-B53 in a canonical diagonal-binding mode with Vα and Vβ domains engaging α2 and α1 domains of B53, respectively[17,19], with a shape complementarity value (Sc)[20] of 0.54. The calculated docking angle relative to the antigenic peptide is 38°, falling into the expected 22°–87° range[17], as shown in the top view of the TCR-binding footprint of QW9-B53 (Fig. 5c). The buried surface area (BSA)

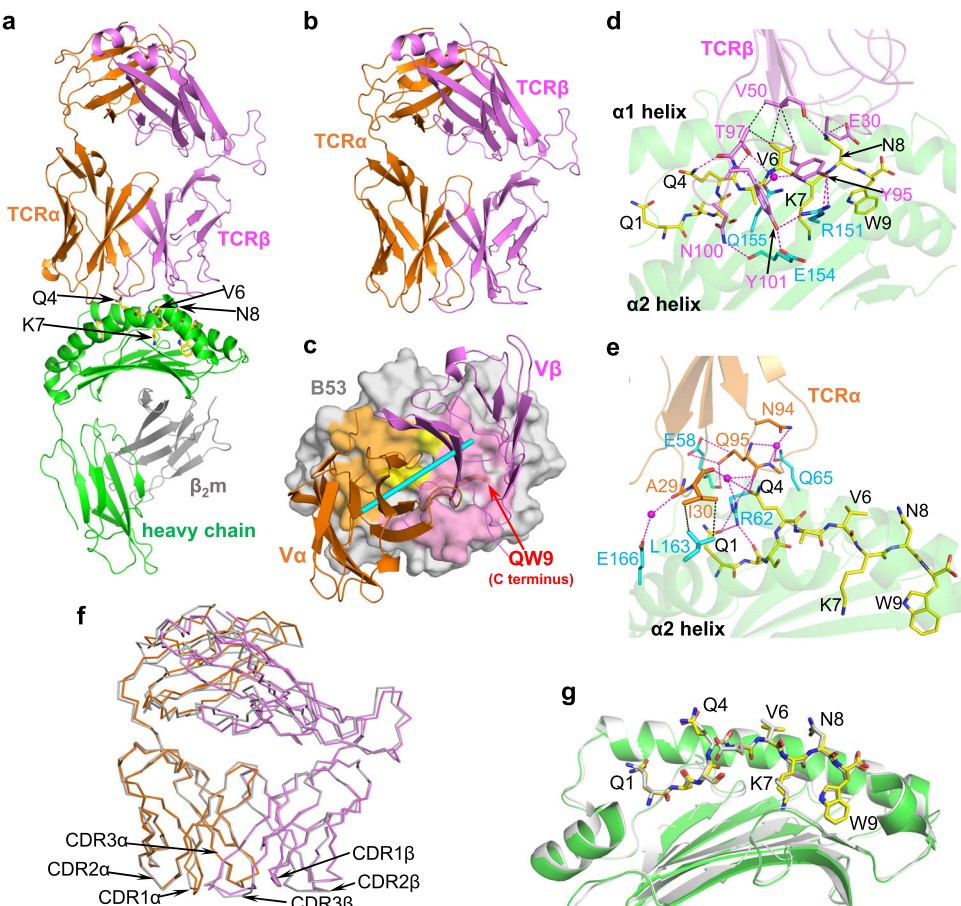

**Fig. 5 | Crystal structures of C3-QW9-B53 and C3. a, b** Crystal structures of the ternary C3-QW9-B53 complex and the apo C3 TCR. TCRα and TCRβ chains of C3 are represented in orange and purple, respectively. The B53 heavy chain (green), light chain (gray), and peptide (yellow stick) are the same as in Fig. 2a, The exposed residues of QW9 for C3 recognition and the buried K7 of QW9 are labeled with arrows. **c** C3 TCR-binding footprint on QW9-B53 surface (gray). The TCRα footprint is light orange; the TCRβ footprint is light magenta. The cyan pole (defined as a line linking the centers of mass of Vα and Vβ domains) shows the C3 docking angle relative to the antigenic peptide. The yellow region represents shared contact residues of QW9-B53 by both TCRα and TCRβ. **d, e** The detailed interactions

between TCRβ (light magenta ribbon) and QW9-B53 (green ribbon) and between TCRα (light orange ribbon) and QW9-B53 (green ribbon) are shown. QW9 is shown as a yellow stick. Residues involved in the interaction are also shown as sticks (TCRα: orange; TCRβ: magenta; B53: cyan). The magenta broken lines represent the hydrogen bonds, while the black broken lines represent hydrophobic contacts. **f** Structural alignment of the apo C3 (gray for both chains) and the complexed C3$^{C3-QW9-B53}$ (TCRα: orange; TCRβ: magenta). **g** Structural alignment of the apo QW9-B53 (gray stick-gray ribbon) and the complexed QW9-B53$^{C3-QW9-B53}$ (yellow stick-green ribbon).

contributed by each CDR loop is listed in Supplementary Table 4. The critical residues involved in C3 TCR interacting with QW9-B53 are summarized in Table 1, while the detailed interactions are shown in Fig. 5d for TCR Vβ binding and Fig. 5e for Vα binding. These extensive hydrogen bonds and hydrophobic interaction ensure recognition specificity and binding affinity. One prominent feature of the C3-QW9-B53 complex is that there is very little conformational change to either C3 TCR or QW9-B53 upon ligation, as seen in the superposed apo form (unconjugated TCR or pHLA) and complex form for ligand-binding Vα and Vβ domains of C3 TCR (Fig. 5f) and QW9-B53 (Fig. 5g).

Upon TCR-pHLA ligation, TCR CDR loops often change conformation, in particular the CDR3α and CDR3β loops[17], but also other CDR loops[21]. However, one important observation revealed by the structure of C3-QW9-B53 is the mostly preconfigured conformation of the CDR loops of the C3 TCR for QW9-B53 binding (Fig. 6a–f). The major change in the complex exists only at CDR1α (RMSD, 1.2 Å) (Fig. 6e) and to a lesser extent at CDR3β (RMSD, 0.68 Å) (Fig. 6f) due to key hydrogen bonds formation (Fig. 6e, f). However, the rest of the CDR loops, even CDR3α (RMSD, 0.38 Å), remain essentially unchanged (Supplementary Table 5). Figure 6a–d shows that it is the internal hydrogen bonds within these loops that largely restrict their flexibility. It is noteworthy that the preconfigured CDR2β loop (Fig. 6d) firmly

assists a β bulge at V50$^{C3β}$ that serves as a major van der Waals contact with V6$^{QW9}$ for critical recognition (Fig. 6g).

Another important observation shown by the ternary structure of C3-QW9-B53 is that the QW9 epitope is "poised" for interaction in B53 with very limited change compared to the structure of the unliganded peptide. An antigenic peptide within the peptide-binding groove of an MHC-I molecule usually protrudes outward toward the C-terminal part, often with intrinsic flexibility, and assumes its defined conformation only upon TCR engagement[22]. Figure 6g illustrates how the C3 TCR reacts with the B53-bound QW9 peptide overlaid with the QW9 from the unliganded QW9-B53 structure. Particularly notable is the fact that sidechains of exposed QW9 residues Q1$^{QW9}$, Q4$^{QW9}$, V6$^{QW9}$, and N8$^{QW9}$ are very well defined by electron density maps (Supplementary Fig. 1b, f) and adopt the same conformation in both apo and complex structures (Figs. 5g, 6g).

It is quite manifest that the sidechains of the three TCR-engaging QW9 residues Q4$^{QW9}$, V6$^{QW9}$ and N8$^{QW9}$ in the liganded QW9-B53 have "poised" their conformations for C3 TCR binding. The residue Q4$^{QW9}$ plays a central part in specific recognition with 5 hydrogen bonds to both Vα and Vβ domains of C3 TCR (Fig. 6g). The V6$^{QW9}$ makes a prominent head-on VDW contact (Fig. 6g) to V50$^{C3β}$ perched on the preconfigured CDR2β loop of C3. The last interacting residue N8$^{QW9}$

**Table 1 | The contacts between C3 TCR and QW9-B*5301**

| Contacts between C3 and B*5301 | | | |
|---|---|---|---|
| CDR loop | TCR residue | HLA residue | Bond type |
| CDR1α | Asp27 | Glu58 | HB |
| | Ala29 | Trp167 | vdw |
| | Ile30 | Leu163 | vdw |
| CDR2α | none | none | none |
| CDR3α | Gln95 | Glu58 | HB |
| | Gln95 | Arg62 | HB,vdw |
| | Al96 | Gln65 | vdw |
| CDR1β | none | none | none |
| CDR2β | Val50 | Thr73 | vdw |
| | Ile54 | Thr73 | vdw |
| CDR3β | Try95 | Arg151 | HB,vdw |
| | Gly98 | Gln155 | HB |
| | Asn100 | Glu154 | HB |
| | Try101 | Arg151 | HB |
| | Try101 | Glu154 | HB |
| Contacts between C3 and QW9 | | | |
| CDR loop | TCR residue | Peptide residue | Bond type |
| CDR1α | none | none | none |
| CDR2α | none | none | none |
| CDR3α | Gln95 | Gln4 | HB |
| CDR1β | Glu30 | Asn8 | HB |
| CDR2β | Val50 | Asn8 | HB |
| | Val50 | Val6 | vdw |
| CDR3β | Thr97 | Gln4 | HB |
| | Thr97 | Val6 | HB,vdw |

Amino acid residues that contribute to the interaction between the TCR and the peptide-HLA complex are listed in the table. The interacting residues are categorized into three groups: the peptide, the HLA molecule, and each individual CDR loop of the TCR. The top section of the table summarizes the pairwise interacting residues between the TCR's CDRs and the HLA molecule. The bottom section of the table summarizes the pairwise interacting residues between the CDRs and the QW9 peptide. The interactions are characterized as either hydrogen bond (HB) or van der Waals (vdw), with the van der Waals cutoff distance being ≤ 4 Å.

forms hydrogen bonds to E30$^{C3β}$ of CDR1β and the mainchain of V50$^{C3β}$ of CDR2β, respectively (Figs. 5d, 6g). Collectively, the above data structurally rationalize the efficiency of C3 TCR binding to the highly networked QW9 peptide presented by the disease-neutral *HLA-B53* allele.

**QW9_S3T backbone conformational changes significantly impede C3 TCR binding**

We next addressed the molecular basis of the reduced QW9-B53-restricted TCR binding of the QW9_S3T variant as indicated by SPR measurements (Fig. 4d, f). Superimposition of the structure of QW9_S3T-B53 onto the complex C3 TCR-QW9-B53 demonstrates that due to the previously described large backbone changes of QW9_S3T at the C-terminal half of QW9, all three TCR-engaging QW9 residues undergo conformational changes, which would be expected to affect TCR binding (Fig. 6h). First, the Q4$^{QW9\_S3T}$ carbonyl group flips away from hydrogen-bonding to T97$^{C3β}$ (Fig. 6h). Due to the V6$^{QW9\_S3T}$ mainchain change the whole residue swings away (Fig. 6h) and can no longer reach V50$^{C3β}$ for VDW contact (Fig. 6g). At the same time, the residue N8$^{QW9}$ cannot maintain the original interaction with E30$^{C3β}$ because the sidechain of E30$^{C3β}$ appears to collide with the outward sidechain of K7$^{QW9\_S3T}$ of the mutant peptide. As mentioned before, the exposed K7$^{QW9\_S3T}$ is actually very much stabilized by extensive internal VDW contacts to the HLA residues V152$^{B53}$ and A150$^{B53}$ (Supplementary Fig. 4b). Therefore, K7$^{QW9\_S3T}$ is likely to change the E30$^{C3β}$

conformation and affect its interaction with N8$^{QW9}$. Given the rigidity of preconfigured CDR loops of C3 TCR, these peptide conformational changes caused by mutation may preclude QW9_S3T cross-recognition by C3 TCR.

To test the above notion regarding the interaction between N8$^{QW9}$ and E30$^{C3β}$ directly, we made two C3 TCRs mutants, C3_E30A$^{C3β}$ and C3_E30G$^{C3β}$, and measured their binding affinity with QW9-B53 via SPR. The $K_D$ value significantly drops from 1.57 μM in wild-type C3 to 44 μM and 73 μM for C3_E30A$^{C3β}$ and C3_E30G$^{C3β}$ mutants, respectively (Fig. 6i). These data provide strong support to show that the C3 TCR-QW9-B53 ternary complex is of biological significance and the dramatic conformational changes of key TCR-binding residues in QW9_S3T impair TCR engagement of QW9_S3T-B53.

**QW9_S3T binds less stably to B53 than to B57**

Antigen presentation by HLA is another important factor for T cell functionality. We further explore the HLA-associated differential CTL recognition of the QW9_S3T variant by experimentally comparing the effect of the S3T mutation on peptide-HLA stability in the context of B53 and B57. Employing differential scanning fluorimetry (DSF)[23], we measured the thermal stability of QW9 and QW9_S3T bound to B53 and B57, respectively. Whereas QW9_S3T-B53 manifests less thermo-stability than QW9-B53 (Fig. 7a), QW9_S3T-B57 is as stable as QW9-B57 (Fig. 7b).

Although QW9 and its S3T mutant have very different conformations, both QW9 and QW9_S3T bind each HLA similarly (Fig. 3c, f). We did observe notable local structural differences in peptide-binding between QW9_S3T-B53 and QW9_S3T-B57 due to the sequence differences in the α1 helix of the two molecules (Fig. 7c, d). In B53, E5$^{QW9\_S3T}$ is tightly sandwiched between R97$^{B53}$ and N70$^{B53}$, meanwhile, N70$^{B53}$ is tightly sandwiched between E5$^{QW9\_S3T}$ and I66$^{B53}$ (Fig. 7e). This tight packing results in CO and NH groups of N70$^{B53}$ being too close to Cγ of E5$^{QW9\_S3T}$ (3.1 Å) and Cγ2 of I66$^{B53}$ (3.5 Å), respectively (Fig. 7c). In contrast, for B57, the long sidechain R97$^{B53}$ is replaced by a short sidechain V97$^{B57}$ (Fig. 7e), and the short hydroxyl group of S70$^{B57}$ forms hydrogen bonds to the mainchain and sidechain of N66$^{B57}$ (Fig. 7d). We hypothesize that the less favorable local environment gives rise to the reduced stability of the QW9_S3T-B53 complex, in large part due to N70$^{B53}$.

To test this hypothesis derived from structural considerations, we generated a mutated HLA construct B53$^{N70S}$ and produced soluble QW9_S3T-B53$^{N70S}$ for additional fluorimetry studies. The N70S mutation in B53 indeed increased the melting temperature (Tm) by 10 °C when bound to QW9_S3T (Fig. 7f), indicating that N70$^{B53}$ is a critical destabilizing factor for QW9_S3T presentation by B53. Conversely, S70$^{B57}$ likely adds to the stability of variant peptide bound to B57 and may contribute to B57-restricted CTL response upon the viral mutation at a polyclonal level. This enhanced variant presentation by B57 may be another key factor to the spontaneous immune control of HIV, consistent with the identification of this critical residue in the genome-wide association study[4] and the latest HLA mutation studies[24]. However, the less stable QW9_S3T-HLA-B53 is still able to elicit corresponding CTLs as we observed in our study (Fig. 2b, d), which suggests that the differential pHLA stability is a contributing factor but might not be the most critical factor determining the differential CTL cross-recognition in terms of these two HLAs presenting QW9 and the variant in our study.

## Discussion

In this report, we address the molecular basis for differential cross-recognition of the mutationally constrained, highly networked antigenic peptide QW9 targeted by HIV-specific CTLs in the context of two *HLA* alleles with differential disease outcome: the disease-neutral HLA-I molecule B53 in comparison with the disease-protective HLA-I molecule B57. The significant differences between the two HLAs include: 1)

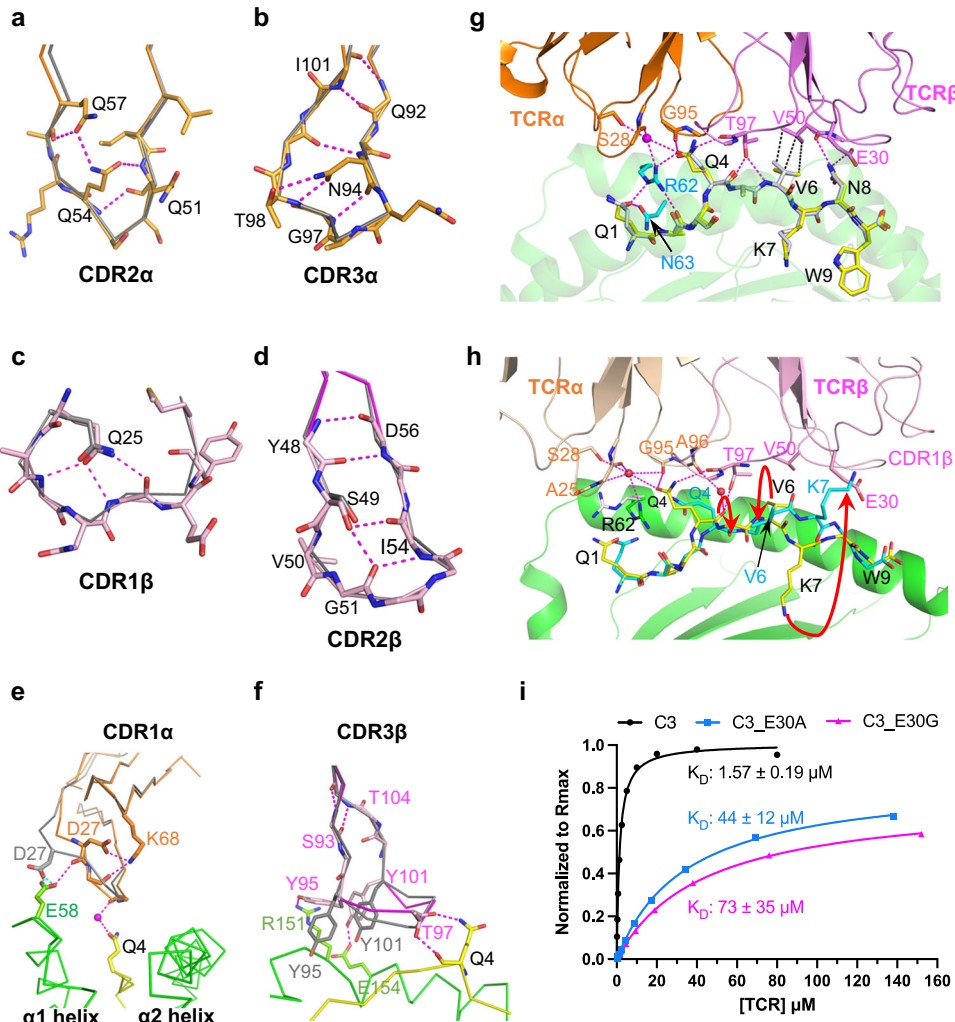

**Fig. 6 | Conformational characteristics of C3-CDR loops. a–d** Internal hydrogen bonds pre-configure conformation of CDR loops (CDR2α, CDR3α, CDR1β, and CDR2β) for ligand binding. Shown are superimposed loops of apo C3 (gray) and the complexed C3$^{C3\text{-}QW9\text{-}B53}$ (CDRα: orange; CDRβ: magenta). The hydrogen bonds are drawn as broken lines. **e, f** Conformational changes of CDR1α and CDR3β upon QW9-B53 binding. CDR1α and CDR3β loops before binding are shown in gray, and after binding in orange and magenta, respectively. QW9 is shown as a yellow stick, and B53 is illustrated as the green skeleton. **g** Superimposition of the apo QW9-B53 and C3-QW9-B53 complex depicts the engagement between TCR and peptide. QW9 from the apo QW9-B53 and C3-QW9-B53 complex are shown as silver and yellow sticks, respectively. The magenta broken lines represent hydrogen bonds, while the black broken lines represent hydrophobic contacts. **h** Overlay of QW9_S3T-B53 onto the C3-QW9-B53 complex. Cyan and yellow sticks depict QW9_S3T and QW9$^{C3\text{-}QW9\text{-}B53}$, respectively. Red arrows indicate the conformational changes from wild-type QW9 to the QW9_S3T mutant. **i** Representative steady state SPR measurements for C3 and C3 mutants interacting with QW9-B53. QW9-B53 was coated on the chip, and refolded TCRs served as the analyte flowed over the chip surface. (Black curve represents wild-type C3; Cyan curve represents C3_E30A; Megenta curve represents C3_E30G; duplicate number, $n = 2$).

the residues R97$^{B53}$ vs. V97$^{B57}$ and N70$^{53}$ vs. S70$^{57}$ buried in the binding groove, which may affect peptide binding; 2) the motif R62$^{B53}$-Q65$^{B53}$ vs. G62$^{B57}$-R65$^{B57}$ exposed on the α1 helix of the HLA, which may directly affect TCR docking. Structural analyses show that both B53 and B57 have similar presentations of the wild-type QW9 peptide as well as a naturally arising QW9_S3T mutant, despite the fact that, compared with QW9, QW9_S3T substantially changes the peptide conformation for both B53 and B57 in a similar fashion. Interestingly, only B57⁺ individuals, but not B53⁺ individuals, exhibited broader recognition of QW9_S3T. Using dual tetramer staining and single-cell sequencing, we found certain distinctive T cells restricted by B57 have better cross-recognition of QW9 and QW9_S3T, implying that a distinct TCR docking against QW9-B57 versus QW9-B53 might function to cross-react. In addition, we also found reduced thermal stability of the variant peptide when complexed with B53 but not with B57, due to specific HLA residue differences between B53 and B57, which were confirmed by HLA mutagenesis studies. Nevertheless, our study

demonstrates that less stable variant peptide bound B53 is still able to elicit CTL responses. Together, these data indicate that minor HLA differences differentially impact TCR recognition of an epitope variant and peptide-HLA stability in the context of a mutationally-constrained, highly networked epitope. Thus, although it may be possible to redirect the immune system to target highly networked epitopes in the context of non-disease protective *HLA* alleles, targeting a naturally arising mutation may be influenced by the restricting *HLA* allele, an important consideration for T-cell-based vaccine design.

A key element of this study is the comparative structural analysis of the same epitope and its mutant bound to disease-neutral B53 and disease-protective B57. Importantly, the structure of the QW9-B53 restricted C3 TCR and the trimolecular structure of C3 TCR complexed with QW9-B53 revealed a nearly rigid-body binding of TCR to pHLA. Notably, even the TCR CDR loops, aside from minor alterations in the CDR1α loop, do not change their conformation, which might endow the binding with an entropic advantage. Such rigidity may preclude

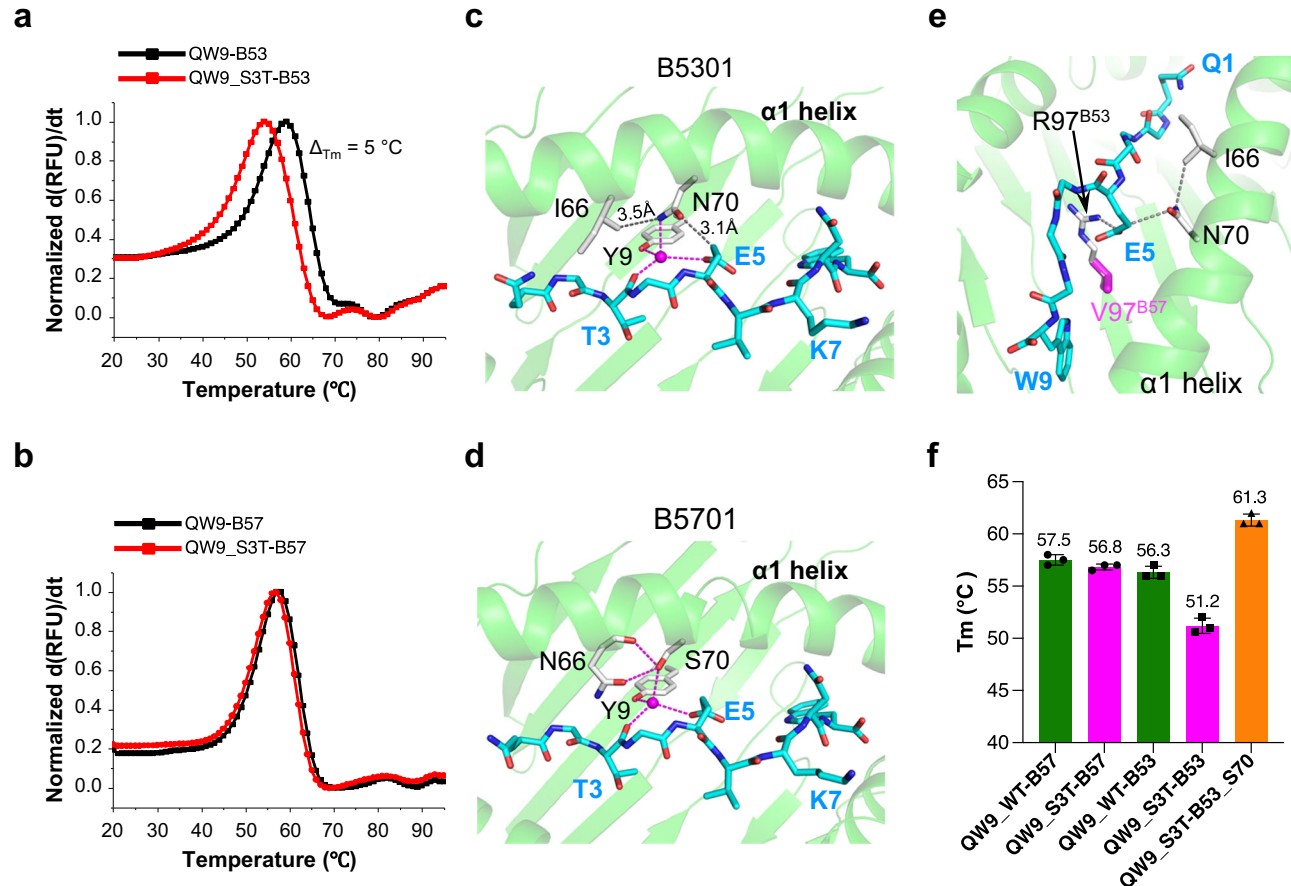

**Fig. 7 | Structure and thermal stability of B53 and B57 presenting QW9 and its mutant. a**, **b** Representative normalized first derivative analysis and associated fit for differential scanning fluorimetry data for B53 (**a**) and B57 (**b**) presenting either QW9 or QW9_S3T peptide. **c**, **d**, Pairwise analysis for QW9_S3T (cyan sticks) for B53 (**c**) versus B57 (**d**). Residues from B53 or B57 are represented as sliver sticks.

Hydrogen bonds are drawn as magenta dashed lines. Contacts between hydrophilic and hydrophobic groups are drawn as gray dashed lines. **e** Sandwich packing model of E5$^{QW9\_S3T}$ in QW9_S3T-B53. **f** Summary of melting temperature (Tm) for QW9 and QW9_S3T bound to wild-type and mutated B53s. The averaged Tm of each tested protein is labeled on the column (Error bars represent SD; duplicate number, $n = 3$).

TCR cross-recognition as we found that the QW9_S3T mutant affects this QW9-B53-specific TCR engagement. Given the observation of some CTLs restricted by B57 having better cross-recognition of QW9_S3T over QW9, it would be interesting to explore the possibly different binding stereochemistry with which a subset of cognate TCR could engage both QW9-B57 and QW9_S3T-B57.

Thermal melt measurements of the peptide-HLA complexes were also performed, providing additional experimental data to suggest that the mutant QW9_S3T is less stable in the context of B53, compared to B57. This may also contribute to the observed differences in cross-reactivity. HLA molecules not only should have an appreciable affinity to bind the targeted peptide but also importantly should be able to retain the bound peptide at the cell surface with sufficient durability (the dwell-time) for a CTL to encounter. Indeed, there is increasing evidence indicating that stability, rather than affinity of pMHC-I complexes, might be better associated with immunogenicity[25,26]. The latest study of SARS-CoV-2 mutations in MHC-I-restricted epitopes evading CD8[+] T cell responses further links pHLA stability to CTL cross-reactivity at a polyclonal level (distinct from the B57 restricted TCR cross-recognition in our study)[27]. A similar conclusion was also reported in an HIV study in which pHLA stability influences CTL immunodominance hierarchies[28]. In our study, we show that an appreciable change in pHLA stability caused by a naturally occurring viral escape mutation is dependent upon HLA polymorphism in just a single residue.

This study does have limitations worth noting. We studied only a single highly networked epitope presented by two different *HLA* alleles but were able to do so in great depth. We were able to generate crystal structures for QW9 and the QW9_S3T mutant for both B53 and B57 and perform thermal melt analyses for both but were only able to assess the ternary TCR-peptide-HLA at a structural level for understanding the weaker cross-recognition of QW9_B53 restricted TCR. A cross-reactive TCR complex with QW9_B57 and/or QW9_S3T-B57 could be valuable for future explorations of the mechanism of T cell cross-recognizing HIV variants. Moreover, it seems that a higher degree of TCR cross-reactivity is related to a higher risk for the development of autoimmune diseases. B57, for instance, has also been associated with autoimmune psoriasis and hypersensitivity reactions[29]. Interestingly, many autoimmune TCRs tend to bind pMHCs near the N-terminal part of antigenic peptide (for instance, see PDB: 1YMM, 1ZGL, 2WBJ, 4P2R). Such an N-terminal eccentric docking might allow the QW9-B57-restricted TCR to avoid the centrally protruding portion of an epitope QW9_S3T, as depicted in a published pMHC structure[30]. The differential residues located at the α1 helix of B57 and B53 defined in our study might modulate different TCR docking. Of note, Q1$^{QW9}$ projects outwardly and freely in QW9_B57 (Fig. 3c) and might offer the putative TCR interaction at the N-terminus. Therefore, future structural studies of different peptides inducing CTL responses in the context of different *HLA* alleles, together with studies of B57-restricted TCR cross-

recognition and functional assays, will be important to expand upon our findings.

In summary, we have exploited the finding that a single highly networked epitope is presented by two HLAs with differential impact on disease course to perform detailed functional, structural and biophysical analyses of recognition of the wild-type peptide and a naturally arising mutant. Despite recognition of the wild-type peptide for both HLAs, we observed diminished cross-reactivity to its variant for disease-neutral B53 compared to disease-protective B57. These data indicate that for a highly mutationally constrained epitope, the restricting *HLA* allele can differentially impact the cross-recognition of naturally occurring variants, with important implications for vaccine design.

## Methods

### Samples and research participants

All samples were obtained from cryopreserved PBMCs from HIV-1-infected individuals with informed consent after study approval by the Mass General Hospital Institutional Review Board (Protocols 2003P001894, 2003P001458 and 2010P002463). Characteristics of the research participants are shown in Supplementary Table 2. Elite controllers were defined as having plasma HIV-1 RNA below the level of detection for the ultrasensitive assay ( < 75 RNA copies/mL by cDNA or <50 copies/mL by ultrasensitive PCR) without antiretroviral therapy and viremic controllers as having HIV-1 RNA < 2000 RNA copies/mL without antiretroviral therapy. CD4[+] T cell counts and viral loads were quantified by standard clinical assays[31]. High-resolution *HLA* genotyping was performed by Dr. Mary Carrington at the National Cancer Institute using sequence-specific PCR in accordance with standard procedures and deposited in Cellular Immunology Database of Ragon Institute of MGH, MIT and Harvard.

### Proliferation assay

PBMCs from the study participants were stained with carboxyfluorescein succinimidyl ester (CFSE, Cat. C34554, ThermoFisher) by incubating the cells with 1 µM CFSE solution (1/1000 (v/v) dilution) for 20 minutes at 37 °C and 5% $CO_2$ and then washed twice in R10 media. The cells were plated at 250,000 cells per well in a 96-well round-bottom plate with 0.5 µg/mL of the corresponding peptides for seven days. The peptides (United Biosystems) used were QW9 (QASQEVKNW) and QW9_S3T (QATQEVKNW). The negative control well had no peptide, and the positive control well had PHA (Cat. 11249738001, Sigma) at 5 µg/mL. On day 7, the cells were stained with IVE/DEAD Fixable Violet dye (Cat. L34964, ThermoFisher) for viability (1/1000(v/v) dilution), Anti-human CD3 BV650 (Cat. 300468, BioLegend) for CD3 (1/100(v/v) dilution), and Anti-human CD8 FITC (Cat.301006, BioLegend) for CD8 (1/100(v/v) dilution). The Cells were analyzed for the percentage of CFSE[low] cells within the CD3[+]CD8[+] singlet lymphocyte gate by flow cytometry.

### Tetramer staining

Biotinylated HLA-B*5701 and HLA-B*5301 monomers refolded with QW9 (QASQEVKNW) or the mutant peptide QW9_S3T (QATQEVKNW) were customed from the company immunAware. Monomers were then tetramerized using phycoerythrin (PE)- or allophycocyanin (APC)-conjugated streptavidin (Cat. 405203; Cat. 405207, Biolegend). Tetramers were validated to rule out non-specific binding with HLA-matched HIV-negative samples. Cryopreserved PBMCs from research participants were incubated with the corresponding QW9 and/or mutant tetramers for 25 minutes at 37 °C and 5% $CO_2$. The cells were then stained by LIVE/DEAD Fixable Violet dye (Cat. L34964, ThermoFisher) for viability (1/1000 (v/v) dilution) and anti-CD3 (Cat. 300468, BioLegend) and anti-CD8 (Cat. 301006, BioLegend) antibodies (1/100 (v/v) dilution) for surface markers and 5 nM corresponding tetramer for TCR. Flow cytometry data were acquired on an

LSR Fortessa (BD Biosciences). Data were analyzed using Flowjo (Tree Star).

### scTCR profiling

TCR repertoire library preparation and sequencing T cell receptor (TCR) beta libraries were prepared from tetramer-sorted cells, 28 cells per subset sorted in 96-well plate, with the SMARTer Human TCR a/b Profiling Kit (Cat. 634434, Takara) following the provider's protocol. The library sequencing was performed on an Illumina MiSeq sequencer (Cat. 634435, Takara) using the 600-cycle MiSeq Reagent Kit v3 (Cat. MS-102-3003, Illumina) with paired-end, 2 ×300 base pair reads. Paired reads containing i5 and i7 indices (Cat. 634435, Takara), plus a third inline index were demultiplexed in 2 steps, first with TaKaRa SMARTer Human scTCR Demultiplexer using the TaKaRa inline index, then with a custom Perl script to demultiplex based on the Illumina i5 and i7 indices. In both cases, a single mismatch was allowed for indices. TCR assembly, identification of *V, D, J* and *C* gene segments, and imputation of full-length *TCR* sequences were then performed using MIxCR version 3.0.13. The *TCRβ* sequence were included in the Source Data files.

### Generation of CD8[+] T cell clones

PBMCs were stained for 20 minutes at 4 °C. Viable, QW9-HLA-tetramer[+], CD3[+], CD8[+] single cells were sorted using a FACSAria II (BD Biosciences) at 70 p.s.i., into each well of 96-well plates, with irradiated allogeneic PBMCs and monoclonal antibody 1ug/ml 12F6 (Cat. ENZ-ABS621-0200, Enzo Life Sciences) to CD3 as a stimulus for T cell proliferation[10] and incubated at 37 °C and 5% CO2. Developing epitope-specific clones were further tested by an IFN-γ ELISpot assay with optimal epitopes and with tetramer staining. Cloned CD8[+] T cells were maintained by restimulation every 14–21 days with monoclonal anti-CD3 and irradiated allogeneic PBMCs in RPMI-1640 (Cat. RNBL4512, Sigma) medium containing 50 U/mL of home-made recombinant IL-2[32].

### Generation of autologous targets cells for killing assays

Autologous Epstein-Barr virus-transformed B cell lines were used as targets in the chromium release assays. Ten million frozen PBMC were thawed and resuspended in 1 mL of RMPI (Cat. RNBL4512, Sigma), 1.5 mL of fetal bovine serum (Cat. 10438-026, FBS), and 1.5 mL of unconcentrated supernatant of Epstein-Barr virus. Cyclosporine A (Cat. C1832, Sigma) was added in a 1 µg/mL concentration. Cells were cultured for 6 to 8 weeks at 37 °C and 5% $CO_2$.

### Generation of QW9-specific T cell effector cells for killing assays

QW9-specific T cell lines (TCL) were generated from each research participant as following: 10% of the total PBMC cells were incubated with QW9 (QASQEVKNW) peptide at a concentration of 5 µg/mL for 1 hour at 37 °C and 5% $CO_2$, washed three times to remove unbound peptides and then transferred to the remaining 90% PBMCs. The cells were added to 96-well round-bottom plates with 50 U/mL of home-made interleukin-2 (IL-2) in 200 µL per well. The plate was incubated at 37 °C and 5% $CO_2$ for seven days, and the TCL was tested by tetramer staining on day 7. Fluorescence-activated cell sorting (FACSAria II BD) was used to sort QW9-HLA-tetramer[+]CD8[+]CD3[+] cells as effectors in chromium release assay.

### Chromium release assay

Target cells were incubated with respective QW9 and QW9 variant peptides at a concentration of 10 µg/mL and incubated with ⁵¹Cr at a concentration of 250µCi/mL for one hour in a 37 °C and 5% $CO_2$ incubator. ⁵¹Cr-labeled cells were washed 3 times in RPMI media with 10% FBS and resuspended at a concentration of 1 million cells/mL. Target cells were plated in a flat-bottom 96-well plate. To ensure an equal number of QW9-specific cells, tetramer-positive cells were sorted to be used in the killing assay and were added at the appropriate

effector-target ratios. Spontaneous and maximum releases were determined by incubating the labeled target cells with medium alone or 5% Triton X-100 (Cat. A16046.AP, ThermoFisher), respectively. The supernatant was collected after 6 hours of incubation at 37 °C in 5% $CO_2$. We used a Perkin Elmer TopCount NXT Microplate Scintillation & Luminescence Counter to measure the radioactivity present in the supernatant. Quantification of specific killing was calculated as specific killing =100 x (sample release – spontaneous release) / (maximum release - spontaneous release).

## Ex vivo ELISpot assay

IFN-γ ELISpot assays were performed according to the manufacturer's instructions with the Human IFN-gamma ELISpotBASIC kit (ALP) (Cat. 3420-2 A, Mabtech). 100 K PBMCs per test were then incubated with each peptide at a final concentration of 20ug/mL overnight. 0.5uM PHA (Cat. 11249738001, Sigma) and 1uM CMV peptides (Cat. 3619-1, Mabtech) were used as positive controls. RPMI-1640 medium with DMSO (Cat. 317275, EMD Millipore) was used as a negative control. To quantify antigen-specific responses, mean spots of the negative control wells were subtracted from the positive wells, and the results were expressed as spot-forming units (SFU).

## Sequencing of C3 T cell receptor

Fluorescence-activated cell sorting (FACSAria II BD) was used to sort QW9-HLA-tetramer[+]CD8[+]CD3[+] cells into RLT buffer with 1% beta-mercaptoethanol (Cat. S M6250, Sigma). RNA was purified with RNeasy Micro Kit (Cat. 74004, QIAGEN) and converted to cDNA with the 5'RACE cDNA amplification kit (Cat. 634858, Takara). Gene-specific amplification was done using nested PCR and primers to the TRBC region. Amplicons were cloned into a topo TA vector for sequencing and One Shot TOP10 Chemically Competent *E. coli* (Cat. C404003, ThermoFisher) were transformed. Bacterial colonies were picked at 24 h. DNA was extracted with Miniprep Kit (Cat. 27104, Qiagen) and sent for Sanger sequencing at the MGH DNA core. IMGT V-QUEST tool was used for the identification of *TCR* gene segments and CDR3 regions. The TCR sequence was included in the Source Data files.

## TCR Null Jurkat generation

Jurkats were transduced w/ Cas9-blastR, selected for blasticidin and sub-cloned, then transduced with a CRISPR guide against *TCRα* (sgTRAC−GAGAGTCTCTCAGCTGGTACA) under zeocin selection and against *TCRβ* (gTRBC−GCGTAGAACTGGACTTGACAG) under puromycin selection, sub-cloned again and then deep sequenced to confirm clonal double knockout of *TCRα* and *TCRβ* genes.

## Transduction and flow cytometry

C3 TCR sequence (see the Source Data) was cloned into the transfer vector as GFP-2A-C3aChain-2A-C3bchain. 293T cells (Cat. CRL-11268, ATCC) were transfected with transfer, envelope (HF-VSVG - Addgene), and packaging (psPax2 - Addgene) vectors in a 1:1:1 molar mass ratio with Lipofectamine 3000 reagent (Cat. L3000150, ThermoFischer) following the manufacturer's recommendations. 293T cells were allowed to incubate for 3 days, and the supernatant containing the prepared lentivirus was collected, filtered, and concentrated with PEG-it (Cat. LV810A-1, System Bio). Concentrated lentiviral solution (400 µL) was added to each well along with 1 mL Jurkat cells (in R10 media, 1e6/mL) and mixed. Cells were spun down by centrifugation at 800 *g* for 1 h at 32 °C and returned to the incubator. After 2 days, 1 mL of R10 media was added to each well, and cells were assayed for GFP expression and TCR expression with standard tetramer staining.

## Statistics for functional assays

For statistical analysis, we used the Prism v7&v8 program from GraphPad Software Inc. Paired *t* test or non-parametric Mann−Whitney tests were used to compare groups as noted on the figure legend of

each experiment. *p*-values ≤ 0.05 were considered statistically significant. Error bars represent standard deviation.

## Surface plasmon resonance experiments

Surface plasmon resonance experiments were performed with a Biacore T200 instrument using Series S sensor SA chip (Cat. BR100531, Cytiva). All experiments were performed in HBS-EP buffer (Cat. BR100826, Cytiva) at 25 °C. Throughout the experiment, C3 TCR was treated as the analyte, and the pMHCs were treated as ligands. C3 TCRs were recombinantly expressed, refolded and purified into HBS-EP buffer as previously described, and the biotinylated pMHCs were obtained from immunAware. For steady state experiments, the ligands: QW9-B53 and QW9_S3T-B53 were immobilized on Series S sensor SA chip, at chip densities of roughly 1000 response units. C3 TCR flowed over all four-flow cells at a flow rate 10 µL/min. TCR-pMHC binding was measured over a series of TCR concentrations ranging from 0 µM to 160 µM. Each TCR injection comprised 200 s contact time followed by 1000 s dissociation time. For multi-cycle kinetic experiments, the ligands- QW9-B53 and QW9_S3T-B53 were immobilized over Series S sensor SA chip, at chip densities of roughly 100 response units. C3 TCR flowed over all flow cells at a flow rate of 50 µL/min, and $k_{on}$, $k_{off}$ rates were measured over a series of analyte concentrations ranging from 0 µM to 60 µM. Each analyte injection consisted of a contact time of 120 s followed by a dissociation time of 700 s. Experimental data were processed in BiaEvaluation 4.1 and were fit to a 1:1 binding model. While calculating the kinetic parameters, $k_{on}$, $k_{off}$ and the $R_{max}$ were fit globally.

## Differential scanning fluorimetry

Differential scanning fluorimetry was performed using Bio-Rad CFX-96 Real-Time PCR. For thermal stability measurements, 20uL of pMHC sample was mixed with 0.2uL of 100X SYPRO orange dye (Cat. 4461146, ThermoFisher). The temperature scan rate was fixed at 1 °C/min, with the temperature range spanning from 20 °C to 95 °C. Data analysis was performed using OriginPro as previously described (Hellman et. al 2016). Apparent Tm values were determined by identifying the point at which the transition was 50% complete.

## Construction and expression of soluble proteins

DNA sequences of the ectodomains of B57 and B53 heavy chains, C3 TCR-α, and -β chains were optimized for the *E. coli* expression system and cloned into pET22b (+) vector (Cat. 69744-3, Novagen), between *NdeI* (Cat. R0111S, NEB) and *XhoI* (Cat. R0146S, NEB) restriction enzyme cleavage sites. Recombinant B57, B53, TCRα, and TCRβ were produced in *E. coli* BL21(DE3) (Cat. C600003, ThermoFisher). A final concentration of 1 mM IPTG (Cat. I56000, RPI) was added to induce expression of inclusion bodies for 4 hours at 37 °C when the growing cell density reached an $OD_{600}$ value equal to 0.8[33]. Cell pellets were suspended and lysed in the extraction buffer (50 mM Tris-HCl (Cat. 46-031-CM, Corning), 100 mM NaCl (Cat. 71382, Sigma), 2% Triton X-100 (Cat. A16046.AP), pH 8.2) with fresh lysozyme (Cat. L6876, Sigma), DNase-I (Cat. 10104159001, Sigma), and PMSF (Cat. 36978, ThermoFisher). After cell lysis via sonication, inclusion bodies were collected at 10000 rpm for 10 min. To sufficiently lyse cells and make inclusion bodies purer, the inclusion bodies were suspended and sonicated one more time. The inclusion bodies were washed 2−3 times by wash buffer (50 mM Tris, 20 mM EDTA, pH 8.0). The construct of β2m, the HLA light chain, was from the Barbara Uchańska-Zieger's lab (Institut für Immungenetik, Charité-Universitätsmedizin Berlin, Freie Universität Berlin, Berlin, Germany). The β2m inclusion bodies were expressed and purified by the same protocol.

## Refolding and purification

A specific peptide is essential for HLA refolding. The peptide QW9 and its variant in our study, QW9_S3T were synthesized by United

BioSystems Inc. To refold the soluble HLA protein, the heavy chain (56 mg), β2m (28 mg), and peptide (10 mg) were diluted in the refolding buffer (100 mM Tris-HCl pH 8.0, 0.4 M arginine (Cat. A5131, Sigma), 0.5 mM oxidized glutathione (Cat. G4376, Sigma), 1.5 mM reduced glutathione (Cat. G4251, Sigma), 2 mM EDTA (Cat. E4884, Sigma), 4 M urea (Cat. U5128, Sigma), 0.2 mM PMSF) in a volume of 500 mL over 24 hours at 4 °C. The refolding solution was then dialyzed several times against 10 mM Tris-HCl pH 8.0 at 4 °C by using a 6-8 kDa molecular mass cut-off dialysis membrane (Cat. 132670, Spectrum). After dialysis, the refolding solution was concentrated to a small volume and loaded onto the superdex75 (GE Health) gel filtration column for molecules separation in the running buffer composed of 10 mM Tris-HCl, 100 mM NaCl at pH 8.0. Then a Mono-Q ion exchange column was used to remove the small amount of incorrectly refolded protein. The superdex200 increase column (GE Health) was applied for buffer exchange. To refold the soluble C3 TCR and its two mutants, C3_E30A$^{C3β}$ and C3_E30G$^{C3β}$, the total amount of 50 mg inclusion bodies of TCRα and TCRβ (molar ratio, 1:1) were diluted in the refolding buffer (100 mM Tris-HCl pH 8.0, 0.4 M arginine, 0.1 mM oxidized glutathione, 1 mM reduced glutathione, 2 mM EDTA, 4 M urea, 0.2 mM PMSF) in a volume of 1 L over 24 hours at 4 °C. The refolding solution dialyzed twice against ddH$_2$O and twice against 10 mM Tris-HCl buffer using the same 6-8 kDa molecular mass cut-off dialysis membrane. The same strategies for the purification of HLA were applied to TCRs. All purifications were performed on ÄKTA pure chromatography system (Cytiva) and analyzed via UNICORN 7.

## Crystallization

All pHLAs and TCR protein samples were concentrated to ~12 mg/mL in the same solution buffer (10 mM Tris-HCl, 100 mM NaCl, pH 8.0). Crystallization Kits from Hampton (Index HT, Cat. HR2-134; Crystal Screen HT, Cat. HR2-130), and Anatrace (Microlytic Top96, Cat. Top96) were selected as initial screen conditions, and the crystals were obtained at room temperature by using the sitting drop vapor diffusion method. The robots NT8 and Rack Imager made by Formulatrix were employed in both crystallization condition screening and optimization. The crystals of QW9-B57 were from 15-20% (w/v) PEG4K, 20% (w/v) 2-propanol, 0.1 M MES, pH 6.5. The crystals of QW9_S3T-B57 were from 30% (w/v) PEG8K, 0.2 M Ammonium sulfate, 0.1 M sodium cacodylate trihydrate pH 6.5. The crystals of QW9_S3T-B53 were from 30% (w/v) PEG8K, 0.2 M sodium acetate, 0.1 M sodium cacodylate trihydrate pH 6.5. The crystals of QW9-B53 were from 30% (w/v) PEG8K, 0.1 M sodium cacodylate trihydrate pH 6.4/6.5 (or HEPES 6.5-7.5). The crystal of C3 TCR formed from 1.0 M Lithium Sulfate, 0.5 M Ammonium Sulfate, 0.1 M Sodium Citrate/Citric acid pH 5.6. The crystal of C3-QW9-B53 complexes was co-crystallized in 21-23% PEG3350, 0.1 M Tris, pH 8.5.

## Data collection, processing, and refinement of crystal structures

Protein crystals were harvested using Mounted CryoLoop (Hampton Research) and flashily frozen in Uni-puck (Hampton Research) soaked in liquid nitrogen. The crystallization buffer plus 15% glycerol was used as the cryoprotectant solution. The diffraction data were collected using 19ID beamline and ADSC Quantum 315 X-ray diffraction detector and SBCCOLLECT tool at APS (Argonne National Laboratories). Diffraction data were indexed, integrated and scaled using the program HKL2000/3000[34] and further processed in CCP4i[35]. To determine the phase of all structures, the molecular replacement was carried out using Phaser of the PHENIX Program Suite[36]. For all pHLAs structure determination, a search model (PDB 1A1M) was selected from the Protein Data Bank. Structure refinement was also performed in PHENIX with XYZ coordinates, real-space, rigid body, individual B-factor, occupancies refinement for all pHLA structures. The CNS refinement was used for the QW9-B57 structure. To determine C3 TCR, we used the TCR structure from a high-resolution pHLA-TCR complex (PDB 1OGA) as a search model, and XYZ coordinates, real-space, rigid body,

individual B-factor, and TLS were selected as strategies for refinement. To determine the C3-QW9-B53 complex structure, the determined QW9-B53 and C3 structures were used as search models for the solution of the complex structure by using Phaser. The initial solution was refined using rigid body refinement first then followed by refinement strategies of XYZ coordinates, real-space, group B-factor, and TLS. All the resulting models were manually inspected and modified with the program COOT[37]. The software PyMol (Copyright (C) Schrödinger, LLC.) was used for structure display and detailed interaction analysis. The statistics of data collection and structure refinement are listed in Supplementary Table 3.

### Reporting summary

Further information on research design is available in the Nature Portfolio Reporting Summary linked to this article.

## Data availability

The data that support this study are available in the article and its Supplementary files or from the corresponding authors upon request. All structural data have been deposited in the Protein Data Bank (https://www.rcsb.org), with PDB codes: 7R7V 7R7W, 7R7X, 7R7Y, 7R7Z, 7R80. Sequence data have been deposited in GenBank BankIt. The accession numbers for the C3 TCRαβ sequence are OQ858871 and OQ858872, the accession numbers for B57-specific TCRβs are OQ858873, OQ858874, OQ858875, OQ858876, OQ858877, and the accession numbers for B53-specific TCRβs are OQ858878, OQ858879, OQ858880, OQ858881. The source data underlying Figs. 1d–g, 2g, 4b, 4d–f, 6i, 7a, b, f are provided in the Source Data file. Source data are provided with this paper.

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

## Acknowledgements

The authors are most grateful to Professor Ian Wilson (The Scripps Research Institute) for his critical reading of early drafts of our manuscript and valuable suggestions. We thank Funsho Ogunshola and Liza Nicole Vecchiarello for sharing reagents for the ELISpot assay and Ryan Jin-Hyung Park for sharing the TCR null Jurkat Cell line and Professors Aaron Schmidt and Daniel Lingwood for sharing the protein purification system and Jiawu Xu for running MiSeq. This work was supported by the Claudia Adams Barr Program for Innovative Cancer Research and internal funds from the Dana-Farber Cancer Institute, a grant from the Ragon Institute and in part by an NIH grant (P01 HL103526) to J.H.W.; by a Ragon Fellowship from Ragon Institute and a scholarship from University of Science and Technology of China to X.L.; by IBM BlueGene Science Program grants (W125859, W1464125, W1464164) to R.Z.; by an HHMI International Student Research Fellowship to P.A.L.; by NIH grants (R01 AI149704 and 5UM1AI144462) to B.D.W.; the Harvard University Center for AIDS Research, an NIH-funded program (P30 AI060354) to B.D.W., which is supported by the following NIH Co-Funding and Participating Institutes and Centers: NIAID, NCI, NICHD, NHLBI, NIDA, NIMH, NIA, NIDDK, NIGMS, NIMHD, FIC, and OAR; the Bill and Melinda Gates Foundation (INV-002703) to B.D.W.; and a gift from the Mark and Lisa Schwartz Family Foundation. The use of SBC 19-ID at Argonne National Laboratory was supported by DOE contract no. DE-AC02- 06CH11357.

## Author contributions

X.L., B.D.W., and J.H.W. conceived the project; X.L., B.D.W., and J.H.W. wrote the manuscript with input from P.A.L., N.K.S., and D.R.C.; X.L. completed the majority of the experiments; X.L. and N.K.S. designed the SPR and DSF experiments; N.K.S., A.Z., O.C.T.M., and S.C. performed the SPR and DSF experiments; P.A.L., P.S., A.P.T., X.L., and D.R.C. performed the immunological experiments; P.A.L., P.S., A.P.T., X.L., D.R.C., and B.D.W. analyzed the functional data. R.N. and A.Z. participated in protein preparation and crystallization; K.T. and J.H.W. participated in X-ray data collection and structure determination of HLAs; S.X. participated in the structure determination of TCR; X.L. and J.M.U. performed the TCR sequencing and data analysis; G.D.G., M.B., T.H., J.K.W., J.C., and R.Z. discussed and contributed to editing the manuscript.

## Competing interests

The authors declare no competing interests.

## Additional information

---

[1]School of Life Sciences, Division of Life Sciences and Medicine, University of Science and Technology of China, Hefei, Anhui 230027, China. [2]Ragon Institute of MGH, MIT and Harvard, Cambridge, MA 02139, USA. [3]Department of Medical Oncology, Dana-Farber Cancer Institute, Harvard Medical School, Boston, MA 02215, USA. [4]Koch Institute for Integrative Cancer Research at MIT, Cambridge, MA 02142, USA. [5]Howard Hughes Medical Institute, Chevy Chase, MD 20815, USA. [6]Structural Biology Center, X-ray Science Division, Advanced Photon Source, Argonne National Laboratory, Lemont, IL 60439, USA. [7]IBM Thomas J. Watson Research Center, Computational Biology Center, Yorktown Heights, NY 10598, USA. [8]Division of Gastroenterology, Massachusetts General Hospital, Boston, MA 02114, USA. [9]Department of Chemistry, Columbia University, New York, NY 10025, USA. [10]Institute of Quantitative Biology, College of Life Sciences, Zhejiang University, Hangzhou, Zhejiang 310058, China. [11]Institute for Medical Engineering and Science and Department of Biology, Massachusetts Institute of Technology, Cambridge, MA 02139, USA. [12]Department of Pediatrics, Harvard Medical School, Boston, MA 02215, USA. [13]Department of Biological Chemistry and Molecular Pharmacology, Harvard Medical School, Boston, MA 02215, USA. ✉e-mail: xiaolong@crystal.harvard.edu; bwalker@mgh.harvard.edu; jwang@crystal.harvard.edu

