## [Peer Review File · Nature Communications]

Molecular basis of differential HLA class I-restricted T cell recognition of a highly networked HIV peptideEditorial note: This manuscript has been previously reviewed at another journal that is not operating a transparent peer review scheme. This document only contains reviewer comments and rebuttal letters for versions considered at *Nature Communications*.

REVIEWER COMMENTS

Reviewer #1 (Remarks to the Author):

The authors have satisfactorily addressed my technical concerns and the study represents a sound and detailed set of structural analyses. However, as previously noted, the results and reported differences in HLA-B epitope recognition are not entirely unexpected, and few new general principles emerge. The relevance to anti-HIV immunity is made as a counterpoint to this point, and has some merit.

Reviewer #2 (Remarks to the Author):

The authors have submitted a revised version of the manuscript. While there have been some changes, they have addressed my major concerns minimally. My first major point addressed mechanism, and I believe the authors have failed to really consider this. I do appreciate it is not my role to draft the manuscript and the authors are of course free to present their data and story in the manner they feel is most appropriate. However, the manuscript still avoids a mechanistic description in favor of considerable data that doesn't really contribute to the story (i.e., the discussion of hydrogen bonds in the discussion - that really doesn't belong there, is not contextualized, and doesn't really say anything about specificity, and the phrase enthalpic advantage is meaningless).

This is summarized by the abstract which states that "dramatic conformational changes of QW9-S3T impaired TCR cross-recognition...." This conclusion is just not supported by the data at hand.

Further, while there has been some inclusion of errors and quantification, there is still no indication of replicates, etc.

Reviewer #3 (Remarks to the Author):

The revised manuscript effectively addresses most points that were raised in the review of the previous version, and also provides appropriate responses to the comments of the other reviewers. The manuscript remains a highly informative exploration of the molecular basis of the HIV "controller" phenomenon.

One point was not fully addressed and may still be worthy of brief consideration. As described in the rebuttal, the authors now mention in the discussion that the rigidity associated with pre-organized CDRs loop likely generates high affinity ("entropic advantage"), but they do not explicitly reference the idea that such rigidity may also preclude QW9 vs. QW9-S3T cross-reactivity, and thus that CDR flexibility may play a role in the controller phenotype, alongside pHLA stability, which is discussed at some length. This is not a make-or-break point, but while it may appear obvious, CDR pre-organization is one of the particularly valuable insights derived from the structure of the C3 TCR in the absence of the pHLA complex, since it may help to explain the ability of the TCR to differentiate between QW9 and QW9_S3T. Furthermore, if CDR flexibility plays an important role in permitting cross-reactivity, is it possible that the stereochemistry of a controller MHC:networked antigen complex may somehow favor CDR flexibility in at least a subset of cognate TCR? If so, such flexibility might also complicate isolation and purification of controller-specific TCR, consistent with the inability of the team to generate QW9-B57-specific TCR yet.

A handful of typographic issues remain:

Line 52: “dramatic*al*”

Lines 52-54: the second half of the sentence is difficult to follow.

Line 101: “...*and* thermal melt analyses,...”?

Line 444” “... a*n* potent TCR.”

Line 474: “Inter*t*estingly...”

Line 490 “...cross-r*activity...”

REVIEWER COMMENTS

Reviewer #1 (Remarks to the Author):

The authors have satisfactorily addressed my technical concerns and the study represents a sound and detailed set of structural analyses. However, as previously noted, the results and reported differences in HLA-B epitope recognition are not entirely unexpected, and few new general principle emerge. The relevance to anti-HIV immunity is made as a counterpoint to this point, and has some merit.

Response: We appreciate the reviewer's assessments and are pleased that his/her concerns have been satisfactorily addressed.

Reviewer #2 (Remarks to the Author):

The authors have submitted a revised version of the manuscript. While there have been some changes, they have addressed my major concerns minimally. My first major point addressed mechanism, and I believe the authors have failed to really consider this. I do appreciate it is not my role to draft the manuscript and the authors are of course free to present their data and story in the manner they feel is most appropriate. However, the manuscript still avoids a mechanistic description in favor of considerable data that doesn't really contribute to the story (i.e., the discussion of hydrogen bonds in the discussion - that really doesn't belong there, is not contextualized, and doesn't really say anything about specificity, and the phrase enthalpic advantage is meaningless). This is summarized by the abstract which states that "dramatic conformational changes of QW9-S3T impaired TCR cross-recognition...." This conclusion is just not supported by the data at hand.

Response: We agree that improved mechanistic discussion will strengthen the manuscript. We have further revised the text in several locations and added new data including the tetramer staining experiment as the reviewer suggested, detailed below, in order to expand and clarify the discussion of mechanisms underlying differential cross-reactivity.

As the reviewer correctly noted in his/her earlier comments, differences in variant cross-recognition cannot be simply explained by differential QW9_S3T peptide structure when presented by HLA-B53 versus HLA-B57. Indeed, our data demonstrate that S3T mutation

changes the structure of peptide presented by both B53 and B57 in a highly similar way (**page 11, lines 20-23; page 12, lines 1-6**). The observed differences in peptide-HLA stability likely represent a contributing factor (**page 18, lines 4-17**), although the difference is relatively small and the relative impact remains unclear, as the reviewer also correctly noted.

In the revised manuscript, we have made efforts to more clearly distinguish cross-recognition of QW9_S3T by QW9-specific T cells, which was different between the two *HLA* alleles, from overall reactivity against QW9_S3T by polyclonal T cell responses (including by QW9_S3T-specific T cells that do not recognize QW9), which was retained for both *HLA* alleles. This distinction should help to avoid potential confusion by readers, contextualize the contributions of peptide-HLA stability, and focus mechanistic discussion on differential TCR cross-recognition.

First, we revised the Abstract (**page 3, lines 8-16**) for improved clarity. The unclear sentence highlighted by Reviewer #3 has been replaced by less ambiguous mechanistic conclusions.

Second, we revised text related to Figure 1 (**page 7, lines 10-17**) to more clearly highlight that these assays measure cross-reactivity of QW9-specific clones against QW9_S3T, which is distinct from the ability of polyclonal T cell responses to recognize QW9_S3T.

Third, we revised Figure 2 and its related text (**page 8, lines 7-23; page 9, lines 1-6; page 19, lines 14-20**) to highlight that T cells are indeed able to mount B53-restricted QW9_S3T-specific responses, despite reduced peptide-MHC (pMHC) stability. These data suggest that differences in cross-reactivity are unlikely to result solely from diminished pMHC stability, although this is likely a contributing determinant based on the data presented.

Fourth, we added new single-cell TCR sequencing data highlighting differences in TCR gene usage between cross-reactive and single-reactive T cells (**Figure 2e, f**), which are also described in the text (**page 8, lines 20-23; page 9, lines 1-6**). These new analyses provide additional mechanistic insight by demonstrating clonotypic segregation between single-reactive and cross-reactive cells, further distinguishing cross-reactivity from overall reactivity and suggesting differential TCR docking modes and/or CDR flexibility as determinants of differential cross-reactivity.

Fifth, as suggested by the reviewer, we added flow cytometric fluorescence intensity data from tetramer titration experiments (**Figure 2g and Extended Data Figure 4**) with duplicates and enough cell numbers, which suggest that TCR binding affinity is similar

between QW9-B57 and QW9_S3T-B57 for those cross-reactive T cells (**stated on page 13, lines 1-4**). The absence of cross-reactive B53-restricted cells is consistent with our biophysical affinity measurements for pHLA-B53 and C3 TCR and further establishes robust TCR cross-recognition of QW9 and QW9_S3T by B57-restricted T cells.

Sixth, we removed the detailed discussion of structural elements in favor of an expanded mechanistic discussion, as suggested by the reviewer (**page 9 lines 8-23; page 10, lines 1-20; page 13, lines 6-23; page 14, lines 1-11; page 20, lines 1-23; page 21, lines 1-5**). This discussion now focuses on the contributions of differential TCR binding and peptide-HLA stability in determining variant cross-reactivity (**page 19, lines 3-23; page 20, lines 1-23; page 21, lines 1-5**), including a potential role for CDR flexibility as suggested by Reviewer #3 (**page 20, lines 2-12**). We also discuss important limitations of the study and remaining questions for future investigation (**page 21, lines 7-23; page 22, lines 1-3**).

As a result of these changes, our structural findings are now more appropriately framed in the context of our experimental results, which we believe has strengthened the work and its impact. We thank the reviewer for his/her constructive comments.

Further, while there has been some inclusion of errors and quantification, there is still no indication of replicates, etc.

Response: We have revised the manuscript to indicate replicates for measurements. We also corrected typographical and grammatical errors. We appreciated the reviewer's thorough review of our manuscript.

Reviewer #3 (Remarks to the Author):

The revised manuscript effectively addresses most points that were raised in the review of the previous version, and also provides appropriate responses to the comments of the other reviewers. The manuscript remains a highly informative exploration of the molecular basis of the HIV "controller" phenomenon.

Response: We appreciate the reviewer's enthusiastic assessments and are pleased that most of their concerns have been satisfactorily addressed.

One point was not fully addressed and may still be worthy of brief consideration. As described in the rebuttal, the authors now mention in the discussion that the rigidity

associated with pre-organized CDRs loop likely generates high affinity ("entropic advantage"), but they do not explicitly reference the idea that such rigidity may also preclude QW9 vs. QW9-S3T cross-reactivity, and thus that CDR flexibility may play a role in the controller phenotype, alongside pHLA stability, which is discussed at some length. This is not a make-or-break point, but while it may appear obvious, CDR pre-organization is one of the particularly valuable insights derived from the structure of the C3 TCR in the absence of the pHLA complex, since it may help to explain the ability of the TCR to differentiate between QW9 and QW9_S3T. Furthermore, if CDR flexibility plays an important role in permitting cross-reactivity, is it possible that the stereochemistry of a controller MHC:networked antigen complex may somehow favor CDR flexibility in at least a subset of cognate TCR? If so, such flexibility might also complicate isolation and purification of controller-specific TCR, consistent with the inability of the team to generate QW9-B57-specific TCR yet.

Response: We thank the reviewer for this suggestion and have included a discussion of this possibility in the revised Results and Discussion (**page 16, lines 14-16; page 20, lines 2-12**). We agree that this is an intriguing notion that adds to the potential impact of our results.

A handful of typographic issues remain:

Line 52: "dramatic*al"

Lines 52-54: the second half of the sentence is difficult to follow.

Line 101: "...*and* thermal melt analyses,..."?

Line 444" "... a*n* potent TCR."

Line 474: "Inter*t*estingly..."

Line 490 "...cross-r*activity..."

Response: Each of these errors has now been corrected. We revised the Abstract (**page 3, lines 8-16**) for improved clarity. We thank the reviewer for the thorough review of our manuscript.

REVIEWERS' COMMENTS

Reviewer #2 (Remarks to the Author):

The authors have addressed my concerns.

Reviewer #3 (Remarks to the Author):

The revised manuscript has addressed my few remaining concerns and has incorporated the one substantive suggestion very appropriately. The manuscript remains a highly informative exploration of the molecular basis of the HIV “controller” phenomenon. While I defer to reviewer #2, it does appear that the authors have made a strong, good-faith effort to provide additional mechanistic insight.

REVIEWER COMMENTS

Reviewer #2 (Remarks to the Author):

The authors have addressed my concerns.

Response: We are pleased that his/her concerns have been satisfactorily addressed. We thank the reviewer for his/her constructive comments.

Reviewer #3 (Remarks to the Author):

The revised manuscript has addressed my few remaining concerns and has incorporated the one substantive suggestion very appropriately. The manuscript remains a highly informative exploration of the molecular basis of the HIV “controller” phenomenon. While I defer to reviewer #2, it does appear that the authors have made a strong, good-faith effort to provide additional mechanistic insight.

Response: We are pleased that all concerns have been satisfactorily addressed. We appreciate the reviewer’s enthusiastic assessments and supports.